# The Structure of Turbulence and mixed-phase Cloud Microphysics in a Highly Supercooled Altocumulus Cloud

Paul A. Barrett[1,2], Alan Blyth[2], Philip R.A. Brown[1], and Steven J. Abel[1]

[1]Met Office, Fitzroy Road, Exeter, EX1 3PB
[2]National Centre for Atmospheric Science, University of Leeds

**Correspondence:** Paul Barrett (paul.barrett@metoffice.gov.uk)

**Abstract.** Observations of vertically resolved turbulence and cloud microphysics in a mixed-phase altocumulus cloud are presented using *in situ* measurements from an instrumented aircraft. The turbulence spectrum is observed to have an increasingly negative skewness with distance below cloud top, suggesting that longwave radiative cooling from the liquid layer cloud is an important source of turbulence kinetic energy. Turbulence measurements are presented from both the liquid cloud layer and ice virga below. Vertical profiles of both bulk and microphysical liquid and ice cloud properties indicate that ice is produced within the liquid cloud layer at a temperature of -30° C. These high resolution *in situ* measurements support previous remotely-sensed observations from both ground based and space borne instruments, and could be used to evaluate numerical model simulations of altocumulus clouds at spatial scales from eddy resolving to global numerical weather prediction models and climate simulations.

## 1 Introduction

Mixed-phase layer clouds are common in the Earth's atmosphere (Zhang et al., 2010; Warren et al., 1988), from the tropics where detrainment from convection forms long-lived altocumulus layers (Stein et al., 2011), to the mid-latitudes where humidity is brought to the mid-troposphere by cyclonic activity (Rauber and Tokay, 1991). Upwards air-motion associated with gravity-waves may also generate altocumulus cells.

Carey et al. (2008) observed that mid-latitude altocumulus layer clouds were of mixed-phase composition on more than two-thirds of occasions and that mixed-phase conditions were observed within a few tens of metres of observable cloud top and extended down through cloud. Peak LWC was found at cloud top, and ice water content (IWC) reached a maximum in the lower half of the cloud system and the similarity to Arctic boundary layer mixed-phase stratocumulus was noted. More than half of the observed clouds are thinner than 500 m, with mean liquid water content (LWC) of 0.14 g m$^{-3}$ (Korolev and Field, 2008), with thinner clouds being correlated with lower temperatures. Fleishauer et al. (2002) found, for altocumulus in the mid-latitudes, that cloud systems can consist of single and multiple layers. The maintenance of altocumulus clouds is the result

of a complex network of processes relating supercooled water to ice through long-wave radiative cooling (LWRC), turbulence, underlying aerosol properties and entrainment, similar to that found in Arctic boundary layer clouds (Morrison et al., 2012).

The glaciation of a liquid cloud has significant consequences for fractional cloud coverage and albedo. A liquid or mixed-phase altocumulus cloud may have large areal coverage and significant optical depth, although the amount of condensed water may be relatively low, with 50 % of clouds having liquid water path (LWP) $\leq 100$ g m$^{-2}$ (Korolev et al., 2007), as observed by *in situ* instrumented aircraft. Radiative transfer calculations performed by Hogan et al. (2003b) suggest that the radiative impact of the liquid layer is extremely significant. Once glaciated the coverage can be much reduced and the optical depth of

the ice cloud much lower and so understanding the processes involved in the production of ice particles is crucial for being able to quantify the radiative balance of the global climate system (Sun and Shine, 1995). Marsham et al. (2006) showed that maintenance of the supercooled liquid layer in a large eddy simulation of mixed-phase altocumulus was dependant on good representation of the distribution of vertical velocity fluctuations as derived from ground based radar and lidar. Previous *in situ* observations show the range of turbulent fluctuations in altocumulus in the UK to be typically $\pm 1$ m s$^{-1}$ and with a root-mean-

square value of 0.5 m s$^{-1}$ in the middle of the (liquid) cloud (Watson, 1967). Similar results were found by Fleishauer et al. (2002). Ansmann et al. (2009) presented ground based remote sensing observations of the life-cycle of a tropical altocumulus cloud system above Cape Verde (T = -34° C), which began as a liquid layer cloud, and prior to the development of ice had vertical velocity fluctuations, w$'$ $\pm$ 1.0 m s$^{-1}$, with a standard deviation of 0.44 m s$^{-1}$. A warmer mid-latitude cloud with T = -6° C observed from the ground by Simmel et al. (2015) had a similar range -1.5 m s$^{-1}$ < w$'$ < 1.0 m s$^{-1}$.

Westbrook and Illingworth (2011) found that the supercooled water in mid-level clouds is drastically underestimated in GCMs (General Circulation Models) and NWP (Numerical Weather Prediction) models. The GCM simulations are found to have too little cloud in the mid-levels, resulting in a warm bias in sea surface temperatures, one of the largest of which is found in the Southern Ocean (Bodas-Salcedo et al., 2014). Observations made by Mason et al. (2014) found that warm-conveyor belt type clouds, characterised by warm advection and moderate to strong vertical ascent, i.e. altocumulus, were partly responsible

for the bias. However, optically thin mid-level clouds were shown to be of low global significance by Hartmann et al. (1992). Systematic GCM deficiencies are often attributed to low vertical resolution in the mid-levels, poor mixed-phase microphysics, and a lack of subgrid-scale processes such as cellular convection (Hogan et al., 2003a; Bodas-Salcedo et al., 2008). The lifetime and albedo of the clouds are found to be extremely sensitive to the properties of aerosols and ice nucleation processes in climate models (Storelvmo et al., 2011).

The problem of resolution is compounded at higher altitudes, where model levels are more widely spaced, as thinner clouds are correlated with lower temperatures (Korolev et al., 2007), something which also makes *in situ* measurements of mid-latitude altocumulus clouds difficult. Many of the existing studies took place before the impact of shattering of ice on cloud physics instrumentation was fully appreciated (Korolev et al., 2011) and so there is some doubt cast on the quality of the microphysical measurements.

The aim of this paper is to report on improved cloud and aerosol microphysical observations from a detailed case-study and place them in the context of new highly detailed turbulence measurements in order to better understand the processes that maintain mixed-phase altocumulus clouds in the mid-latitudes. The results should also be relevant to mixed-phase layer

clouds in other geographical locations, but similar temperature regimes. The following section details the instrumentation and methods of data processing. A case study of altocumulus is presented in section 3. Results are presented in section 4, including the thermodynamics and turbulence structure, and cloud microphysics followed by discussion and conclusions.

## 2   The Case Study

A mid-latitude cyclone was centred off the southern tip of Greenland on $2^{nd}$ February 2012. The warm front extended eastwards across Iceland towards Scandinavia, whilst the cold front extended roughly north-south over the Atlantic Ocean a few hundred kilometres west of Ireland. The broad warm sector covered the north-west of the UK with high pressure to the south and east. Figure 1 shows a 10.8 $\mu$m infra-red satellite image of the North Atlantic region from AVHRR (Advanced Very High Resolution Radiometer) for the north-east Atlantic including the western UK and Ireland, Iceland and part of Greenland. Extensive layer cloud was observed within the warm sector. Close to the low pressure centre, west of 10 W, the cloud top temperature was colder than -50° C and away from here to the south-east the cloud top temperature was between -18° C and -29° C as estimated using Met Office products derived from MSG (Meteosat Second Generation) and NWP output. Discussion regarding this technique is in Hamann et al. (2014).

*In situ* data were collected from the FAAM BAe146 Atmospheric Research Aircraft (FAAM, 2017) from single-layer mixed-phase altocumulus clouds within the warm-sector of a mid-latitude cyclone, sampled on $2^{nd}$ February 2012 (FAAM Flight B674). Data collection began in the afternoon (1600 UTC) with the end of the measurement period being flown in twilight conditions (1900 UTC). The choice of flight track was restricted by air traffic considerations and so it was impossible to advect with the cloud in a Lagrangian sampling strategy. Fortunately the north-south flight track was closely aligned to the direction of the mean wind, which was predominantly from the south and ranged in strength from 6 m s$^{-1}$ at the southern end of the flight track to 8 m s$^{-1}$ in the north. There was some degree of wind shear above the cloud containing layer, with mean wind direction being close to northwestly. This shear may have resulted in production of turbulence in the layer below.

An optically opaque liquid cloud with cellular structure and an areal coverage of between 6 and 7 Oktas was observed upon arrival into the area, along with ice virga which extended below the liquid cloud base, as shown in the photograph in Fig. 2. Visual observation from the flight deck and real-time inspection of *in situ* data revealed that the cloud-containing layer was capped by a weak temperature inversion ($\approx$ 1 K) below which sat the liquid cloud layer. Liquid cloud top sloped from 5800 m (T = -31° C) at the southern extent of the sampling region, to 5400 m (T = -27° C) at the northern end. Constant altitude legs interspersed with slant profiles were flown between 4500 m and 7500 m. The slant profiles indicated a pseudo-adiabatic liquid water structure with larger liquid water content towards cloud tops, similar to observations by Korolev et al. (2007).

## 3   Instrumentation and Methods

The BAe146 carries a scientific payload capable of measuring meteorological and thermodynamic conditions and bulk and microphysical cloud properties described in part by Mirza et al. (2016) and Allen et al. (2014).

Data were selected from times when the aircraft was flying a constant heading using the limits for rate of change of heading and roll in Table 1. Slant Profiles were flown at a vertical rate of change of 5 m s$^{-1}$, whereas a Straight and Level Run (SLR) had a rate of change of altitude less than 0.5 m s$^{-1}$. During sampling the aircraft typically had a nominal Indicated Airspeed (*IAS*) of 210 kts, approximately 140 m s$^{-1}$ in the mid-troposphere.

Turbulent wind components were sampled in three dimensions at 32 Hz using a 5-port turbulence probe located on the nose of the aircraft (e.g. Petersen and Renfrew (2009)). No problems due to icing of the turbulence probe pressure ports were observed during data collection.

Temperature measurements were recorded at 32 Hz by a non-de-iced Goodrich Type 102 platinum-resistance thermometer and reported at 1 Hz. There was no evidence of contamination on this sensor housing due to the presence of condensed liquid water when compared against the deiced sensor. Humidity data were provided at 0.4 Hz by a near-infrared Tunable Diode Laser (TDL) absolute humidity spectrometer, a WVSS2 (Water Vapour Sensing System Mk. 2) fitted to a flush-mounted inlet (Vance et al., 2014), allowing computation of dew point temperature and water vapour mixing ratio, $q_v$.

Bulk LWC measurements were made using a Nevzorov hot-wire probe (Abel et al., 2014). Data from the Nevzorov total-water sensor (LWC + IWC) were also interrogated to identify regions of cloud-free air that would permit use of out-of-cloud aerosol particle observations; for details see Appendix A1. A Lyman-alpha total water content ($q_t$) instrument (Brown and Francis, 1995) was employed to estimate total water mixing ratio, $q_t = q_v + LWC + IWC$.

Liquid cloud particle size and number concentration were measured using a CDP (Cloud Droplet Probe) with anti-shatter tips fitted (Lance et al., 2010). The performance of the probe was monitored using glass spheres of known diameter using the method presented in Rosenberg et al. (2012). The CDP was calibrated prior to the field campaign and additional glass bead checks were routinely performed throughout in order to ensure that there was no drift in performance. The integrated LWC from CDP compared well with that measured by the Nevzorov hot-wire probe (Pearson correlation coefficient = 0.98), with slight over-reading at larger sizes, where the calibration is more uncertain. Ice particle number concentrations were measured by two optical array probes (OAP), the Cloud Imaging Probes (CIP), with the CIP15 for diameters 30 $\mu$m $\leq$ D $\leq$ 960 $\mu$m in 15 $\mu$m increments and CIP100 (100 $\mu$m $\leq$ D $\leq$ 6.2 mm) in 100 $\mu$m increments (Cotton et al., 2013). Data were processed using the SODA2 package (System for OAP Data Analysis version 2 (Bansemar, 2016)) from NCAR (National Centre for Atmospheric Research). Maximum observed ice crystal diameters were often less than 1 mm in this study and so the impact of shattered ice fragments on microphysical measurements is expected to be low and careful inspection of particle imagery from the CIP probes supports this. Additionally, an algorithm to remove shattering artefacts based on the inter-arrival times of particles was applied. Imagery of cloud particles was provided by a Cloud Particle Imager (CPI) probe (Connolly et al., 2007). Cloud fraction was computed for both liquid (CDP) and ice particle (CIP15) observations by examination of the ratio of 1 second periods with and without cloud particles, in a given altitude range. Full details are given in Appendix A1.

Aerosol particle number concentration and size were sampled in the nominal size range 0.1 $\mu$m $\leq$ D $\leq$ 3.0 $\mu$m with a Passive Cavity Aerosol Spectrometer Probe (PCASP) Knollenberg (1970), which was calibrated at FAAM using the method of Rosenberg et al. (2012). Full details of the data processing are given in Appendix A1.

In the absence of an Ice Nucleus Counter (INC) an assessment of the ice nucleating particle (INP) concentration was made by measuring the number concentration of aerosols with diameter, $D \geq 0.5 \ \mu$m and applying empirically-based temperature-dependent parametrisations from DeMott et al. (2015) for general aerosol particles, and Tobo et al. (2013) for the case of forest emissions of INP.

When flying above cloud the cloud top height (CTH) was measured using data from a downward facing Leosphere ALS450 backscatter lidar, using an integration time of 2 s and vertical resolution of 1.5 m (Allen et al., 2014; Osborne et al., 2014). During a preliminary measurement leg, flying above cloud, it was observed that the cloud top height sloped along the flight track. This, coupled with the fact that air-traffic restrictions prevented Lagrangian cloud sampling in the horizontal. It was therefore decided to develop an air-relative coordinate system, anchored to cloud top height (or inversion altitude) in the vertical, and neglecting the horizontal dimension. Cloud top was constrained at an isentropic surface by a thermodynamic inversion which was crossed 9 times between 1710 UTC and 1735 UTC and a further 6 times later in the measurement period between 1825 UTC and 1840 UTC, in the southern half of the region. Many of these profiles contained cloud. The cloud layer below the inversion was identified from measurements of total water mixing ratio. The cloud layer had $q_t = 0.725$ g m$^{-3}$, as compared to the free troposphere above, where $q_t = 0.45$ g m$^{-3}$. Within the cloud layer wind speed was greater in the north, and a linear fit was produced to give wind speed as a function of latitude. This allowed for an estimate of *air-relative* horizontal coordinate and hence two new vertical coordinates: the vertical position of the aircraft relative to the spatially and temporally varying i) CTH: $\Delta z_{(CTH)}$ and ii) the inversion (top) altitude: $\Delta z_{(inv.)}$. Estimated time series of both inversion and cloud-top altitude using these spot measurements were generated by interpolating in space and time throughout the measurement period.

To enable use of wind data from slant profiles the three-dimensional wind components were filtered using a 4-pole Butterworth high-pass filter, following the studies of Lenschow et al. (1988) and others (e.g. Mahrt (1985), Brooks et al. (2003)) who have used this method for analysis of stable boundary layers. All data from both slant profiles and geometrically level flight segments were included, outside of turns. Investigation of the length-scales of turbulence in the system were investigated by applying a range of Butterworth filters (see Appendix A2) with filter lengths between 1.5 km and 16 km (Table A1). Turbulence kinetic energy (TKE) was computed from the residual high-frequency fluctuations in three dimensions, for each length of filter, using

$$TKE = \frac{1}{2}\sqrt{u'^2 + v'^2 + w'^2}. \tag{1}$$

Application of Taylor's frozen-turbulence hypothesis (Stull, 1997) permits the compositing of data from throughout the measurement period on to a "virtual met-mast", floating in the free troposphere.

## 4 Results

Figure 3 shows the flight profile on a latitude-altitude cross section. The potential temperature ($\theta$) measurements, although not uniformly distributed throughout the section, were interpolated in latitude-altitude space to give an overview of the thermodynamic environment in which the cloud was found. The liquid water content from the CDP, and ice number concentration ($N_I$)

from the CIP15 are over plotted and show the spatial cellular structure to the liquid water and vertical structure of the cloud system.

The CTH derived from profile observations as a function of time and latitude was extrapolated back in time to allow comparison with the lidar-derived CTH observations from the earlier above-cloud run. Residual differences between the derived time series of CTH and lidar CTH were typically less than 50 m and in all cases less than $\approx$ 100 m (see Table 2). De-trending the lidar-CTH data at two scales, 30 km and 3 km, shows that the range of CTH was of similar order to the uncertainty. The standard deviation of CTH at the 30 km scale was of the order 25 m and at 3 km scale was close to 12 m, with the two scales perhaps corresponding to the altocumulus cell scale (3 km) and gravity wave structures (30 km).

With distance from inversion altitude as the vertical coordinate it can be seen that there was an inversion in potential temperature of the order 3° C which extended over less than 100 m (Fig. 4). Below the inversion was a seemingly well-mixed layer in terms of potential temperature, for at least 600 m below the inversion. The air was much drier above the inversion where relative humidity w.r.t. liquid fell from saturation at the top of the layer to $\approx$ 50 % above (Fig. 4 (c)). Ice supersaturation extended down through the top 500 m of the cloud layer. The liquid cloud at the top of the layer had a depth of just over 200 m with a maximum coverage of 0.8 just below the inversion (Fig. 4 (d)). Ice cloud fraction was close to zero at the top of the liquid cloud layer, increasing with depth below this, to a similar value close to 0.8, some 300 m below cloud top.

The current generation of operational NWP models have a typical vertical resolution of the order 600 m in the mid-troposphere (Walters et al., 2017). If the cloud fraction ($cf$) observations were to be volume averaged over the full depth of the cloud layer, $cf^{grid}$ (similar to a typical NWP grid box depth) then the volume cloud fractions of the liquid and ice parts of the cloud are $cf^{grid}_{liquid} \leq 0.25$, $cf^{grid}_{ice} \leq 0.60$. The resulting mixed-phase cloud fraction is therefore $cf^{grid}_{mixed} \leq 0.15$. The maximum value assumes that both liquid and ice are evenly distributed throughout the volume, something the observations do not support: liquid is concentrated in the top third of the cloud layer, and ice is concentrated in the lower two thirds. This greatly reduces the volume in which liquid and ice are expected to co-exist, which has important consequences for mixed-phase microphysical processes in NWP simulations. Microphysical aspects of the cloud are discussed in detail in Sect. 4.2.

In the following section the analysis begins to focus on the vertical structure of the observed vertical velocity fluctuations in the cloud layer system described above.

## 4.1 Vertical Profiles of TKE and Vertical Velocity Distribution

Figure 5 shows profiles of vertical wind fluctuations, w′ (a) and TKE (b) constructed from raw data with a high-pass Butterworth filter of length 9 km (see Appendix A2). The percentiles are shown in 20 vertical levels with each representing 25 km of measurements. Vertical velocity fluctuations (Fig. 5 (a)) ranged between -3 m s$^{-1}$ and +2 m s$^{-1}$, and the variability, characterised by the standard deviation of w′, increased from inversion top, to a maximum of 0.5 m s$^{-1}$ at -150 m, and remained reasonably constant to -400 m, before diminishing by -600 m. The skewness of the distribution of vertical velocity fluctuations is increasingly negative with distance below cloud top, as the spectra of w′ becomes increasingly dominated by downdraughts. The TKE profile (Fig. 5 (b)) show median values close to 0.5 m$^2$ s$^{-2}$ in a layer from -150 m to -400 m, below which they reduce. Horizontal high-pass filtered wind components, u′, v′, have larger magnitude residual mean values close to the inversion

compared to w′, a result of slight wind-shear across the inversion and contamination by the filter. This will result in spuriously large values for TKE in this region. By substituting the vertical velocity fluctuations, w′, in place of u′ and v′ in the calculation of TKE (Eq. 1) it can be seen that the turbulence kinetic energy tends towards zero, and increases steadily below cloud top towards the local maxima at -150 m. Below this altitude the estimate of TKE using only w′ is biased high as compared to the full three dimensional estimate.

Vertical velocity fluctuation PDFs as a function of distance from inversion altitude are presented in Fig. 6 for five vertical ranges. The highest level, within cloud-tops, centred on $\Delta z_{(inv.)}$ = -50 m, had a narrow w′ distribution, with low skewness, -0.23, and a range that was predominantly $\pm 1.0$ m s$^{-1}$. Within the bulk of the liquid cloud, centred on -150 m the range was larger with peak updraughts of 1.5 m s$^{-1}$ offset by stronger maximum downdraughts of -2.0 m s$^{-1}$, resulting in low skewness of -0.20. Peak updraughts were weaker in the next vertical level down, centred on -250 m and roughly corresponding to the cloud base layer and start of sub-cloud virga, where negative skewness developed to -0.62. Deep within the virga layer at -400 m there will still moderate updraughts generated, but the magnitude of the skewness was largest at -0.79, relaxing to -0.44 at -550 m. The PDF at the lowest level showed most fluctuations were of low magnitude, similar to cloud top, but that occasional strong downdraughts up to -1.5 m s$^{-1}$ were present, still influenced by LWRC from cloud top.

The impact of filter length on the measurements is explored to provide information on the length scales that are operating within the cloud system (Fig. 7). Close to cloud top at -50 m both TKE (standard deviation) and skewness are low, and vary little as a function of length scale. Within the liquid cloud layer at -150 m the TKE is observed to peak, and is dominated by longer length scale circulations. There is low magnitude negative skewness to the distribution of vertical velocity fluctuation, consistent for all filter lengths: here the induced updraughts are almost sufficient to offset the negative skewness introduced by the downdraughts. Just below the liquid cloud is where a separation of skewness as a function of length scale becomes apparent, whilst TKE is generally similar to the level above. By 250 m below cloud top the magnitude of the skewness has doubled in general, with a slightly stronger increase for longer filter lengths. Below -400 m the downdraughts begin to dominate the spectra, with the magnitude of skewness with a 9 km being 50 % larger than for a 1.5 km filter, although the overall magnitude of turbulence intensity is tending to reduce. The lower magnitude skewness for shorter length scales at this altitude implies that resultant updraughts occur at smaller scales than the downdraughts driving the circulation. Updraughts are occurring on a scale of the order a few hundred metres, compared to a couple of kilometres for downdraughts. The strongest difference between long and short filter lengths is apparent at -500 m, although the overall magnitudes of skewness are lower, implying that the few downdraughts that penetrate this low do not have as large an impact in driving updraughts, possibly due to reduced convergence between ever less frequent cold-pools to this depth. Turbulence intensity is a factor of four lower than the peak.

### 4.2   Cloud and Aerosol Microphysical Observations

The updraughts are responsible for the maintenance of the supercooled liquid cloud layer. The liquid water content profile as measured by the Nevzorov LWC sensor is shown in Fig. 8 with data plotted with respect to $\Delta z_{(CTH)}$. The data show a sub-adiabatic profile of liquid water content for the majority of observations, but with the extreme largest values being close to adiabatic.

Theoretical, undilute adiabatic, LWC profiles were calculated by assuming an ascent of a saturated air parcel from three initial altitudes corresponding to potential cloud bases. The first, from the minimum liquid cloud base at -223 m show that

peak observed cloud top LWC values compare well with this theoretical estimate. An ascent from $-167$ m peaks close to the $75^{th}$ percentile of cloud top LWC. A third ascent from -97 m has a peak LWC close to the $50^{th}$ percentile at cloud top. Whilst entrainment of dry air from aloft at the inversion may be non-zero, these calculations demonstrate that the non-uniform cloud base may have contributed to the observed in-cloud variability in LWC at a given level. This suggests a range of turbulent eddies and updraught depths contributed to the overall spectrum of in-cloud liquid water contents.

Now the full mixed-phase cloud system is considered. Statistics were calculated for vertically resolved liquid and ice particle in-cloud number concentrations (Fig. 9 (a)), effective radius (Fig. 9 (b)) and condensed water content (Fig. 9 (c)). Vertical layers of ice supersaturation frequency are shown to aid interpretation.

Cloud drop numbers were constant in-cloud at $\approx 30$ cm$^{-3}$, implying nucleation at cloud base (Fig. 9 (a)). Cloud-top liquid particle effective radius was typically observed in the range 11 $\mu$m to 15 $\mu$m (Inter-quartile range), with smaller particles close

to cloud base (Fig. 9 (b)). As previously shown (Fig. 8) largest LWC values were at cloud top peaking close to 0.1 g m$^{-3}$. Liquid cloud particle size distributions were as expected, with cloud particles growing in size towards cloud tops (Fig. 10). These composite distributions, segregated by distance from cloud top (Table 3) show that clouds bases typically have high number concentrations of small particles, below the mode of $\approx$11 $\mu$m, and very few particles larger than 20 $\mu$m. Within the mid cloud layer there are greater concentrations of larger particles and a similar mode. At cloud tops the largest particles are

about 30 $\mu$m and concentrations of particles smaller than 10 $\mu$m are reduced compared to the layers below, whilst the mode remains constant.

Median ice particle number concentrations measured by CIP15 were of the order 0.5 $L^{-1}$ to 5 $L^{-1}$, peaking just below the liquid cloud base, and remaining approximately constant below this, whilst reducing with altitude towards the top of the liquid cloud (Fig. 9 (a)). Data from CIP100 showed a similar trend, with a peak concentration just below cloud base, with the

245 concentrations falling off more rapidly towards cloud top than data from the CIP15, where particles tend to be smaller. Ice particle effective (Fig 9 (b) - shown for CIP100 - which has a larger sample volume) increased with depth from liquid cloud top, to as maximum value close to 300 m below cloud top. Peak IWC was slightly greater than peak LWC (Fig. 9 (c)), whilst median values were lower in magnitude. Ice particles were not found at the very top of the liquid layer and their concentration peaked in magnitude below the liquid cloud bases. Ice supersaturation frequency (Fig. 9) fell below 100 % at a level of 350 m

below cloud top. IWC reduced below 600 m where ice supersaturation frequency fell below 25 %. Effective diameter decreased below 500 m, where the ice supersaturation frequency was less than 50 %. Ice properties remained consistent, even when ice supersaturation frequency reduced, suggesting that the ice particles initially fell within branches of descending air with larger values of ice supersaturation.

Observations of ice number, effective radius and IWC are consistent with ice nucleation occurring within the liquid cloud.

Following this, ice particles continue to grow and sediment through the ice supersaturated layers beneath. Mean ice particle size distributions are plotted in Fig. 10 (b) according to the vertical levels in Table 3 to show the properties of the ice virga. Number concentrations of the smallest particles are much larger for the very top level in cloud, perhaps identifying the ice

particle generation zone. Below this cloud top layer, the cumulative particle volume distribution has a constant slope up to a size of 300 $\mu$m, implying that the same process is responsible for the ice production - diffusional growth. Particle imagery from CPI (Fig. 10 (b)) showed the dominant ice particle type to be complex polycrystalline structures (also observed by Korolev et al. (2000)) which likely grew as single crystals, followed by aggregation to larger sizes. The particles share many features of "assemblages of large plates" that are present at temperatures close to -30° C above water saturation (see Figure 5 in Bailey and Hallett (2009)). Some evidence of riming was present within the liquid layer

The ice production rate for these altocumulus clouds was calculated in a manner similar to Harris-Hobbs and Cooper (1987), with the addition that the rate was computed between adjacent pairs of size channels (which have a width of 15 $\mu$m) for CIP15 data, up to 400 $\mu$m (full details in Appendix A3). Data were selected from mixed-phase regions ($N_d > 5$ cm$^{-3}$, $N_i > 0.1$ L$^{-1}$ for particles with diameters larger than 100 $\mu$m) for the initial cloud sampling period between 1700 UTC and 1730 UTC and in the three vertical levels defined in Table 3. The results are shown in Fig. 11. Peak rates are notably greater for the level nearest cloud top, and for sizes smaller than 140 $\mu$m. At this cloud-top level there are very little data for sizes larger than 230 $\mu$m. For the two lower levels within the liquid cloud there is general similarity between the computed ice production rates. A summary of these results can be found in Table 3 which presents weighted mean ice production rates for groupings of adjacent size channels on CIP15, in the size ranges 60 – 105 $\mu$m, 120 – 225 $\mu$m and 240 – 345 $\mu$m. For the smallest particles (D < 105 $\mu$m), the rate of ice production near cloud top is double that at the level below, and factor of three larger than at cloud base - which supports the suggestion that ice nucleation occurs at cloud top.

As mentioned in Sect. 3, ice nucleating particles were not observed directly. Total aerosol number concentration (accumulation mode) and concentrations of the larger particles are shown in Fig. 12 (a) for two layers, one above cloud, and one within the cloud layer, when out of cloud/precipitation. The observed number concentrations of aerosol particles larger than 0.5 $\mu$m from PCASP (Fig. 12 (a)) were used as inputs into the parametrisations of DeMott et al. (2015) for general aerosol and (Tobo et al., 2013) for forest sources of aerosol (Fig. 12 (b)). Back trajectories (not shown) indicate a source region for the airmass over the boreal forests of Northern America with large-scale ascent in the mid-latitude storm-tracks providing a potential uplift mechanism. A nucleation temperature of -30° C was applied, assuming nucleation occurred at cloud top. Number concentrations of aerosol particles larger than 0.5 $\mu$m are of the order 0.07 cm$^{-3}$, both above and within the cloud layer. Resulting INP concentrations are $N_{INP} = 1.0$ L$^{-1}$ (DeMott et al., 2015) and 0.60 L$^{-1}$ (Tobo et al., 2013). Observed ice number concentrations are plotted in Fig. 12 (c) from CIP15 and CIP100. Observed ice concentrations compare closely with the INC corrected INP concentrations, particularly for the CIP15 which is able to observe smaller ice particles, and particularly for the forest emissions parametrisation of Tobo et al. (2013). Whilst the observations show that there are larger concentrations of accumulation mode aerosols within the cloud layer than above the same cannot be said for number concentrations of aerosols larger than 0.5 $\mu$m, limiting the ability of the data in this study to distinguish the source of the INP particles. Peak accumulation mode aerosol particle number concentrations are below 10 cm$^{-3}$, and so some particles smaller than 0.1 $\mu$m must have acted as CCN (cloud condensation nuclei) to produce a cloud with droplet concentrations of 30 cm$^{-3}$.

## 5   Discussion

The measurements presented show a highly supercoooled altocumulus layer cloud, with precipitating ice virga. Along with previously published results from the literature, the life-cycle of altocumulus clouds will now be considered. Supersaturation and hence liquid cloud formation in the mid-levels of the troposphere may be achieved through large scale ascent with upwards air motion accentuated through wind shear or gravity wave activity, or convective detrainment at a stable interface (Rauber and Tokay, 1991) (Fig. 13 (a)), where the distribution of relative humidity with respect to liquid permits. Ansmann et al. (2009) showed that for tropical altocumulus clouds the liquid phase is always present before the ice phase, and Westbrook and Illingworth (2011) found that for mid-latitude clouds observed from the ground that liquid cloud layers are situated above ice clouds in the majority of cases warmer than -27° C.

Once the formation of liquid has occurred, cooling from cloud-top through emission of long-wave radiation to space will generate negatively buoyant air parcels that will tend to descend through the cloud layer. Descending air-parcels occur in narrow downdraughts and through mass-continuity force upward motions that, dependent on humidity, may continue to support the production of supersaturation and hence liquid water. A more detailed discussion is presented below. Figure 8 showed that the observed liquid water content followed a pseudo-adiabatic profile: see schematic in Fig. 13 (c)).

At some later stage, and where the liquid clouds reside at temperatures colder than 0° C, the production of ice may occur (Fig. 13 (d)). The computation of ice production rate suggests a tendency towards ice production close to cloud top (Fig. 11), as depicted in Fig. 13 (d-i), but the increase of both ice number concentration and ice mass with distance below cloud top (Fig. 9, (a), (c)) indicates that the production of ice occurs at all levels within the liquid cloud. Number concentrations of ice particles remain constant in the ice-supersaturated layer below the liquid cloud (Fig. 9 (a)) indicating that ice nucleation is not active in that region (Fig. 13 (d-ii)). The negatively skewed vertical velocity distribution (Fig. 6) is similar to that found in nocturnal stratocumulus (Nicholls, 1989). Hogan et al. (2009) (Fig. 13 (b)) found a similar profile of skewness for LWRC driven nocturnal stratocumulus clouds using ground based measurements, in contrast with the profiles of skewness that were obtained when surface heating driven cumulus clouds were overhead. The turbulence kinetic energy spectra that is also similar to that found in stratocumulus (Ghate et al., 2014)) (Fig. 13 (e)). The eventual dissipation of the altocumulus clouds may occur through erosion of humidity within the cloud layer through precipitation, or an increase in subsidence, which would act to warm the layer (Larson et al., 2006). Likewise turbulent mixing and radiation can also be important.

The turbulence observations presented in Sect. 4.1 will now be placed into context, by comparing them with the remote-sensing observations and derived conceptual-model of Schmidt et al. (2014) of the circulation structures within altocumulus clouds. Other observations from ground based studies conducted in both the tropics and mid-latitudes, as well as previously published *in situ* observations will be drawn upon where appropriate. At the top of the cloud, where $\Delta z_{(inv.)}$ = -50 m (Fig. 6), these *in situ* observations show a narrow distribution of $w'$, with a low skewness value, features which are likely to be characteristic of shallow cloud top eddies, similar to those resolved by the remote sensing observations in *Figure 7* of Schmidt et al. (2014): at this level, close to the inversion, the broader-deeper circulations do not have the opportunity to develop, as depicted in Fig. 13 (b-i).

The PDF of w′ at $\Delta z_{(inv.)}$ = -150 m has positive and negative fluctuations of almost equal magnitude, hence low skewness (Fig. 6), and therefore corresponds to the Schmidt et al. (2014) "cloud-layer" where Rayleigh-Bénard type-cells were observed (Fig. 13 (b-ii)). Schmidt et al. (2014) also find a symmetrical distribution of vertical velocities, on a horizontal scale comparable or just larger than the liquid cloud depth. The previous *in situ* observations of turbulence in altocumulus clouds (Watson, 1967; Fleishauer et al., 2002) were made within the liquid layer and therefore all correspond to these liquid layer observations, and show similar magnitudes. Ansmann et al. (2009); Simmel et al. (2015) both presented remote-sensing results for the liquid portion of altocumulus clouds with similar magnitudes and distributions.

Altocumulus clouds often occur in multiple layers (Korolev and Field, 2008; Fleishauer et al., 2002). These could form when the resulting updraughts produce liquid supersaturation at a level below the cloud layer driving the turbulence (Fig. 13 (b-iii)). It is possible that long-lived altocumulus clouds such as those observed by Westbrook and Illingworth (2013) and Carey et al. (2008) were maintained through a particular set of conditions, including vertical stability, and the distribution of humidity, that permitted the formation of secondary "daughter-cells" at the same altitude as the driving clouds.

Within the ice virga, between -250 m and -400 m below cloud top (Fig. 6), the observed turbulence properties differed from those found in the liquid cloud layer with a shift to increasingly negative skewness; these results are consistent with the "Subcloud Layer 1" of Schmidt et al. (2014), corresponding to Fig. 13 (b-iv), with downwards penetrating mammatus type features. This connection between the liquid cloud layer and below was observed by Schmidt et al. (2014) to be on a longer length scale than the in-cloud circulations, of the order 1 km. Ansmann et al. (2009) presented turbulence observation from within the ice virga layer (using radar: sensitive to the larger ice particles) for late-stage development clouds, which showed a shift to negative skewness: $-1.75 \text{ m s}^{-1} < \text{w}^{'} < 0.75 \text{ m s}^{-1}$, and an increase in standard deviation of $\text{w}^{'}$, which are comparable to the direct turbulence observations presented here. Unlike remote-sensing derived observations of turbulence, these *in situ* measurements are not reliant on, or biased by, observations of, or assumptions about the ice particle size distribution. At the lowest level of the *in situ* observations ($\Delta z_{(inv.)}$ = -500 m, Fig. 6), the magnitude of both skewness and TKE is lower, and the width of the PDF of w′ is as narrow as at cloud top, sharing characteristics with the "Subcloud Layer 2" of Schmidt et al. (2014) (Fig. 13 (b-v)).

There is observational evidence in the literature of a link between turbulence and the production of ice in supercooled layer clouds. For example Heymsfield et al. (1991) found that ice production in two altocumulus clouds, both at similar temperatures to each other and to the clouds in this study, differed along with the magnitude of turbulence, and liquid water content. One of the clouds had peak LWC = 0.05 g m$^{-3}$ with vertical velocity values of $\pm$ 0.75 m s$^{-1}$, and produced virga with IWC = 0.08 g m$^{-3}$. The second cloud had peak LWC = 0.02 g m$^{-3}$, and negligible ice concentrations, whilst the vertical velocity fluctuations were < 0.25 m s$^{-1}$. Crucially that cloud with ice had liquid particles of 15 $\mu$m at cloud top, whereas the low LWC ice-free cloud had higher drop concentrations and a mean liquid particle diameter of less than 6 $\mu$m.

Hobbs and Rangno (1985) saw ice enhancement in Arctic stratiform clouds close cloud tops when liquid cloud particles were larger than $\approx$ 20 $\mu$m. Table 3 presents integrated number concentrations of liquid cloud drops, $N_D$, larger than three particular diameters, close to 20 $\mu$m for the three vertical cloud levels. For a particular minimum size the concentrations are greater towards the top of the cloud along with the observed ice production rate. There is a steep gradient in concentrations as

a function of size at each level in this size range, in the case of the top cloud level, up to three orders of magnitude across the three size bins. If indeed the production of ice is dependent on the presence of "large" liquid drops then it could be particularly sensitive to the size distribution of liquid particles. Taking the cloud top maximum ice production rate of 1.6 m$^{-3}$ s$^{-1}$, and numbers of large drops given in Table 3, the production of ice could be sustained over 14 hours assuming that the droplets larger than 18.8 $\mu$m form ice. The steep slope of the PSD in these clouds results in a strong sensitivity to this calculation of ice production timescale. Only close to 2 hours of ice production could be maintained for ice forming droplets larger than 19.8 $\mu$m, yet only 11 minutes if ice formation required liquid particles larger than 21.9 $\mu$m.

Lance et al. (2011) later showed that the CCN budget was capable of modulating the ice phase, but rejected the notion that the inhibition of freezing of small drops due to dissolved solute (noted by de Boer et al. (2010)) was important in the clouds, as the cloud particles typically exceeded 10 $\mu$m, thus negating the effect. Hobbs and Rangno (1985) postulated that the ice may have been formed during partial evaporation of cloud droplets during mixing of dry air from aloft. Later work (Durrant and Shaw, 2005) defined such a process as Contact Nucleation Inside Out (CNIO), similar to the INP production through evaporation mechanism that Fridlind et al. (2007) demonstrated that it could explain ice production in Arctic mixed-phase clouds in eddy-permitting cloud simulations. Increased turbulence resulting from additional cooling in higher-LWC clouds may promote saturation and evaporation cycling of cloud and aerosol particles, which would lead to an increase in the rate of CNIO events for a given temperature range. This mechanism may be able to explain many of the supercooled layer cloud observations including those presented here.

Many of the *in situ* and remote sensing observations discussed above show similar properties to the turbulence structure, seemingly independent of temperature, and geographical location. Given that there is a great variety of altocumulus clouds (Korolev, 2007; Fleishauer et al., 2002), this is somewhat surprising, and so additional observations and studies using eddy permitting simulations should be used to determine the turbulence budget of altocumulus clouds as a function of parameters including temperature, LWP, and CCN. Such studies could then inform representation of, or parameterisation of weakly-forced turbulent layer clouds in NWP and climate models.

New instrumentation including holographic imaging probes may prove useful in trying to identify the conditions and locations in which the ice first forms in liquid layer clouds. Observations are required that can positively identify small ice ($\geq$ 10 $\mu$m) in the presence of large liquid drops ($\approx$ 25 $\mu$m), or that could identify small ice at cloud edges in regions of evaporation and entrainment. Supercooled stratocumulus clouds may provide a more accessible natural laboratory.

## 6  Conclusions

Highly supercooled single layer mixed-phase mid-level altocumulus clouds with a cloud top temperature of -30° C, with precipitating ice virga were measured by an instrumented aircraft. There have been few detailed observational studies of this kind mainly because the clouds are usually transient in nature. The *in situ* observations presented in this paper show the turbulence structure and microphysical properties of this cloud system at high vertical resolution.

Cloud top was found to slope significantly and so cloud top and inversion height were estimated as a function of time and space This reduced the problem to a single vertical dimension allowing the aircraft data to be used as though from a "synthetic met-mast" observing the cloud to advect past. Comparison of thermodynamic and cloud microphysics data from both geometrically level flight segments and slant profiles was therefore possible, increasing the quantity of data available. These observations are a snapshot in time of a dynamic environment and there is an implicit assumption that the turbulence is frozen in an extension of Taylor's Hypothesis to these airborne measurements (Stull, 1997). Application of the same technique to existing layer cloud flight data may be useful. The method was accurate to a scale of approximately 30 m in the vertical, as confirmed by above cloud lidar observations.

The key features observed are turbulence sustained by long wave radiative cooling, resulting in a vertical velocity distribution where negative skewness increases with distance below cloud top, and a PDF in the range $\pm 2$ m s$^{-1}$, but typically $\pm 1$ m s$^{-1}$. Direct measurement of the turbulence in the ice virga layer below the supercooled liquid layer cloud are presented for the first time. These *in situ* observations support the observations-based conceptual-model of turbulence in altocumulus shown by Schmidt et al. (2014), as well as other remotely-sensed ground based observations of turbulence (Simmel et al., 2015) and the cloud life-cycle observations of Ansmann et al. (2009). The measurements are also similar to previous *in situ* observations made in the liquid layers of altocumulus clouds (Watson, 1967; Fleishauer et al., 2002).

The adiabatic-type liquid water content profiles was shown to be consistent with updraughts from variable cloud bases, consistent with turbulence driven from cloud top. The deepest liquid cells were just over 200 m in depth. Cloud drop number concentration was constant with height at 30 cm$^{-3}$, with the modal and maximum size increasing with altitude, suggesting little entrainment of dry air from aloft. Production of ice within the liquid cloud layer was shown to occur at a rate of $0.84 \pm 0.14$ m$^{-3}$ s$^{-1}$ in the middle of the liquid cloud, with evidence that freezing events may be enhanced close to cloud top by a factor of 2 or more to $1.60 \pm 0.48$ m$^{-3}$ s$^{-1}$. Ice particle concentrations appear to be well represented by INP concentrations derived from *in situ* observations of aerosol particle size distributions and parametrisations of DeMott et al. (2015) and Tobo et al. (2013).

*Code and data availability.* Data from the FAAM research flights are available from the Centre for Environmental Data Analysis, www.ceda.ac.uk/badc/fa. Code for processing of core data from FAAM research flights is available from https://github.com/ncasuk/decades-pp. Other bespoke proscessing code is available on request.

## Appendix A: Appendix

### A1 Processing of data from microphysics probes

Optical array probe data from the CIP15 and CIP100 were processed using SODA 2 software (Bansemar, 2016). Specific SODA2 settings used were i) to not reconstruct particles that are at the edge of the diode array, and ii) not apply corrections for out-of-focus liquid drops. OAP probes occasionally suffer from "stuck bits" where one pixel remains constantly "on".

Inspection of the imagery did not indicate that this problem occurred for any of the pixels and so the option to correct for "stuck bits" was turned off. Counts per bin were corrected for size-dependent depth of field and ice particle number concentration calculated using measured true airspeed (TAS) and

$$N_i = \left( \frac{\sum_{bin=1}^{n_{chn}} count(bin)}{DoF(Fn(bin))} \right) /(SA \times TAS). \tag{A1}$$

The errors in sample volume can be up to 100 % for the smallest size bins. To reduce counting uncertainty individuals bins were combined at larger sizes: CIP15: 100 to 420 $\mu$m (2 bins), 435 to 600 $\mu$m (4 bins), > 615 $\mu$m (8 bins), CIP100: 700 to 2800 $\mu$m (2 bins), 2900 to 4000 $\mu$m (4 bins), > 4100 $\mu$m (8 bins). CIP15 data and the CIP100 data agree well where sizes overlap. IWC was calculated from the integrated size distribution using the mass-dimensional relation from Brown and Francis (1995).

Following Ryder et al. (2013) the data from bins adjacent to gain-stage cross-overs in the PCASP were summed into a single, wider, bin, namely bins 4 and 5, and bins 15 and 16. The lowest size channel was rejected as the lower bound is unknown. A composite error was calculated by combining fractional errors from each contributor: bin size, counts and flow rate. It is apparent that, even for low bin counts, the major contribution to the uncertainty in PCASP was from the sizing.

PCASP aerosol data are only valid when out of cloud due to contamination by break up of cloud and precipitation particles. Cloud free regions were determined using the standard deviation of raw-power on the Nevzorov total water content (TWC) probe (TWC = LWC + IWC), exploiting the observation that variability in-cloud is very different from that out-of-cloud. A 1 Hz time series of the standard deviation of electrical power was computed from the 32 Hz data record, and a threshold of 2.0 mW ($\approx 1 \times 10^{-4}$ g m$^{-3}$) placed on this parameter to partition the data into cloud-contaminated and clear-sky time periods with a 2 s "safety window", to account for in-cloud variability, potential timing offsets between individual data-logging system clocks, and cloud edges where the condensed water content may fall below the sensitivity of the probe. Typical out-of-cloud variability was between 0.5 mW and 3 mW, and when cloud microphysics probes reported cloud particles the Nevzorov TWC reported orders of magnitude greater variability, of up to 1 W.

## A2   Butterworth Filters

Turbulent fluctuations within the planetary boundary layer are typically calculated using Reynolds decomposition (French et al. (2007), Petersen and Renfrew (2009)) from vertical stacks of level flight legs by removing a linear trend from the data record. Cloud boundaries of mid-tropospheric layer clouds tend to follow isentropic surfaces, removed for the surface of the Earth. This leads to difficulty when attempting to use this filtering method in altocumulus cloud layers, as geometrically level flight legs are effectively slant profiles through thermodynamic space. Here the high frequency fluctuations horizontal and vertical wind parameters were extracted from the data record by filtering using high-pass 4-pole Butterworth filters, and taking data from both level flight and slant profiles.

A range of filter lengths were applied, to investigate scales of motion within the cloud system (Table A1). It was assumed that TAS = 140 m s$^{-1}$ throughout the measurement period as the true value varied by less than 5 %.

Assuming that the synoptic and turbulence scales are spectrally distinct, a perfect high-pass filter would result in the mean of the parameter in question being equal to zero at all locations. Throughout the depth of the layer the residual mean of the high-pass filtered vertical velocity fluctuations is $\leq 0.01$ m s$^{-1}$ for filter lengths below 5 km, and only as large as $\leq 0.02$ m s$^{-1}$ for a 12 km filter length. The greatest magnitude for any given filter occurred close to the inversion altitude, as a result of smoothing of the vertical wind shear across the inversion, thus contaminating the residual turbulence fluctuations.

Loss of variance (TKE) occurs for filter lengths of up to 9 km: the resolved TKE falls below 90 % of the maximum value for filter lengths shorter than 9 km, below 75 % at 6 km, and below 50 % at 2.5 km. The reduction in overall TKE indicates that scales of this order were contributing to the turbulent fluxes in the vicinity. As a compromise, the 9 km filter was taken to be sufficient to separate the turbulence scales of motion from synoptic and other larger scales of motion, and is set as the default filter.

### A3   Calculation of Ice Production Rate

Calculations of ice production rate from observations were first performed by Harris-Hobbs and Cooper (1987) (hereafter HHC87), originally to calculate the rime splintering secondary ice production (SIP) rate in cumulus clouds within the Hallett-Mossop (HM) temperature range (Hallett and Mossop, 1974), and have been since repeated by Taylor et al. (2016). Those same calculations were applied here, to measure primary ice production rate (PIP), in these mixed-phase altocumulus layer clouds.

HHC87 calculated the ice production rate, P, from the difference in the cumulative size distribution of measured ice particles, C, between two size thresholds, $L_1$ and $L_2$,

$$P = [C(L_2) - C(L_1)]/t_{21}, \tag{A2}$$

where the growth time is given by

$$t_{21} = (L_2 - L_1)/G(T), \tag{A3}$$

and G(T) is the average ice particle growth rate. Using the growth curve from Bailey and Hallett (2012) for water-saturated "Region A" (temperature dependent growth rate for plate-like particles) and the measured layer temperatures (Table 3) the growth rates were found to be of the order $G(R) = 0.38 \pm 0.02$ $\mu$m s$^{-1}$. It was assumed that the growing ice particles remained in a water saturated environment and that in the early stages growth through riming and aggregation was negligible and diffusional growth dominated. Ice particles were assumed to have negligible size upon production as the maximum liquid cloud particles had diameter $\leq 30$ $\mu$m.

Only ice particles larger than 137 $\mu$m were considered in HHC87, but with the current generation of shadow imaging probes, and the maximum cloud droplet size in these altocumulus clouds of only 30 $\mu$m, this restriction can be reduced to smaller sizes. The ice production rate was calculated for particles with diameters larger than 30 $\mu$m and smaller than 400 $\mu$m.

In order to validate this extension to the technique, the same method was applied to calculations of ice production in two cumulus clouds, at T = -5° C and T = -8° C previously sampled by Abel et al. (2017) on 24 November 2013. Both clouds were in the SIP temperature range. Liquid particles as large as 100 $\mu$m were observed, and so calculations considered only

ice particles sizes larger than this, and smaller than 200 $\mu$m. Weighted mean values are $P^{[-5^\circ C]} = 31.44 \pm 4.05$ m$^{-3}$s$^{-1}$ and $P^{[-8^\circ C]} = 11.86 \pm 2.38$ m$^{-3}$s$^{-1}$, more than an order of magnitude larger than that found in these altocumulus clouds, and comparable to previous measurements had by HHC87 and Taylor et al. (2016) (Fig. A1).

For the altocumulus clouds the computed rate is a lower bound because aggregation would act to reduce the observed rate, as would growth through riming, which was observed on some CPI particle imagery. Erfani and Mitchell (2017) present results from previous observational and modelling studies that indicate a range of minimum riming dimension thresholds, $D_{thresh}$, for various ice particle habits, between 35 $\mu$m for hexagonal columns, and 200 $\mu$m for broad-branched plates.

*Author contributions.* PB designed and conducted the research flight and analysed the data and wrote the paper. AB and PRAB were involved
in the development of the analysis methods. AB, PRAB and SJA reviewed the written document.

*Competing interests.* The authors declare they have no competing interests

*Acknowledgements.* Airborne data were obtained using the BAe146-301 Atmospheric Research Aircraft operated by Directflight Ltd and managed by the FAAM, which is a joint entity of NERC and the Met Office. The staff of the Met Office, FAAM, Directflight Ltd, Avalon Engineering and BAE Systems are thanked for their work that made the research flights a success. Franco Marenco (Met Office) is thanked
for his assistance in providing the cloud top height product from the lidar data. James Dorsey (University of Manchester) is thanked for providing the data and imagery from the CPI probe. We thank two anonymous reviewers for their comments which helped us to improve the manuscript.

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

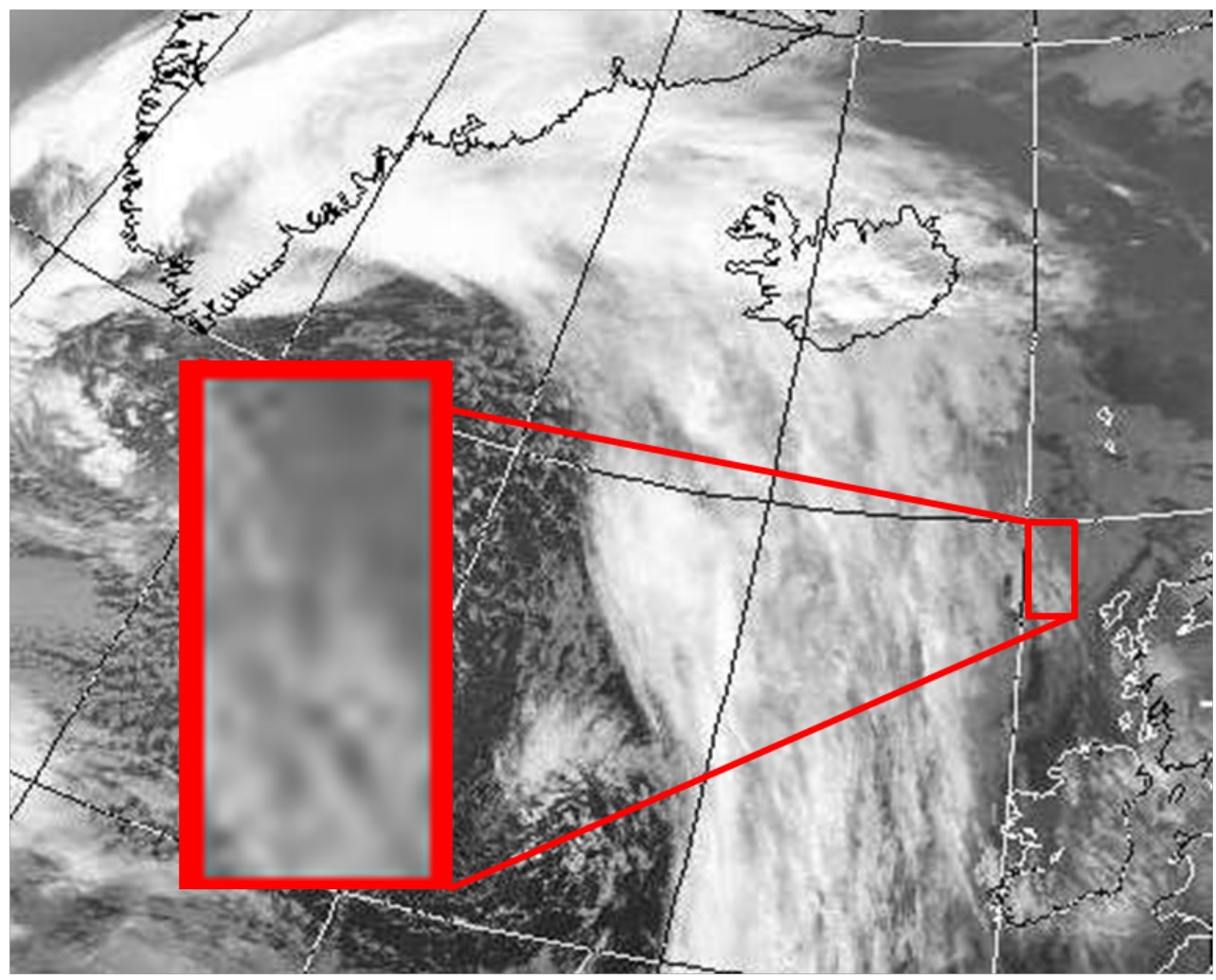

**Figure 1.** Infra-red satellite imagery (10.8 $\mu$m) on $2^{nd}$ February 2012 from AVHRR at 1440 UTC. Layer cloud associated with the warm sector is visible to the west of the UK. Red box [inset] indicates the location of the airborne sampling.

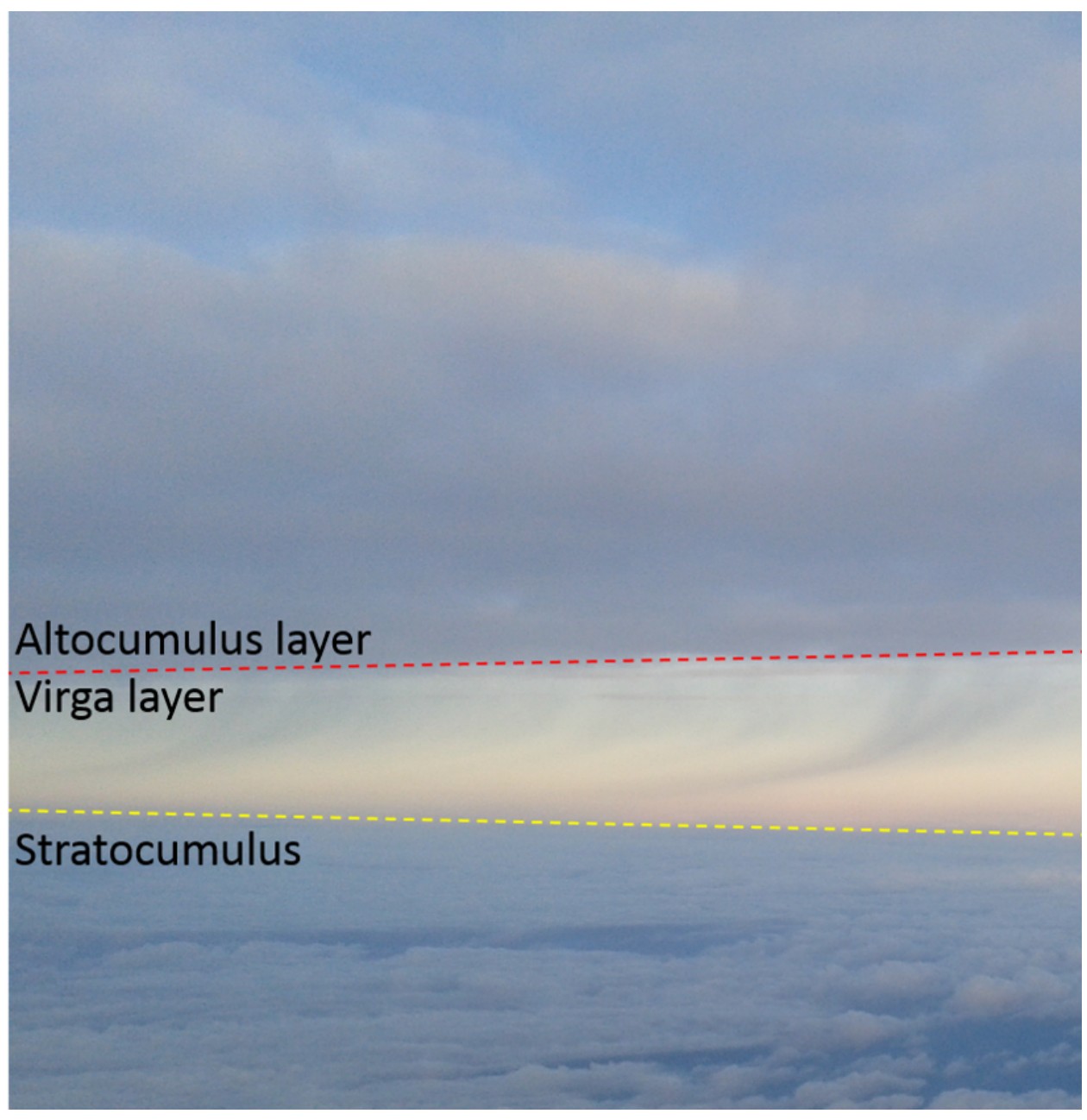

**Figure 2.** Photograph of the Altocumulus layer from underneath the liquid layer, within the precipitating ice virga layer, taken on arrival in the operating area at 1617 UTC, $2^{nd}$ February 2012. The broken cellular structure of the liquid altocumulus layer cloud is visible above the red dashed line. Ice virga can be seen below the liquid cloud, being advected with the mean wind from right to left (south to north), between the red and yellow dashed lines. Boundary layer stratocumulus cloud is visible below the yellow dashed line. *Photo: S Abel.*

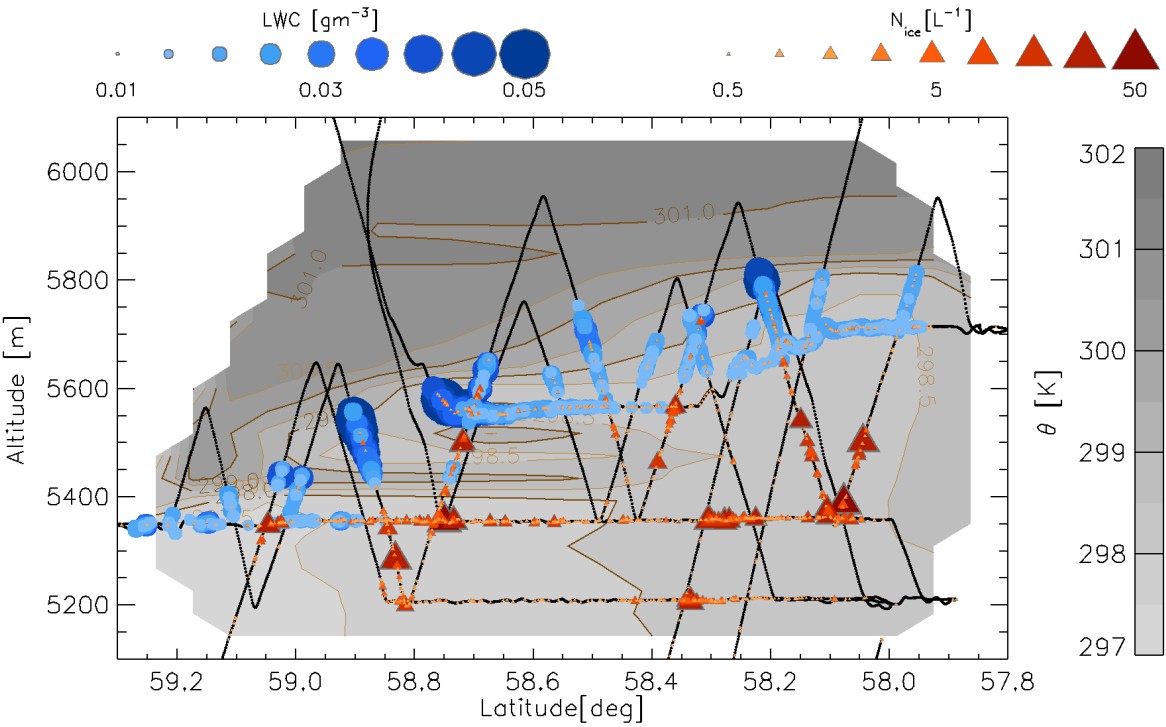

**Figure 3.** Aircraft altitude as a function of latitude from flight B674, $2^{nd}$ February 2012 (black line), LWC from CDP (blue circles - key is top left) and $N_I$ from CIP15 (red triangles - key is top right). Potential temperature ($\theta$) observations, interpolated in altitude-latitude space (grey-scale contours - key is right-hand-side).

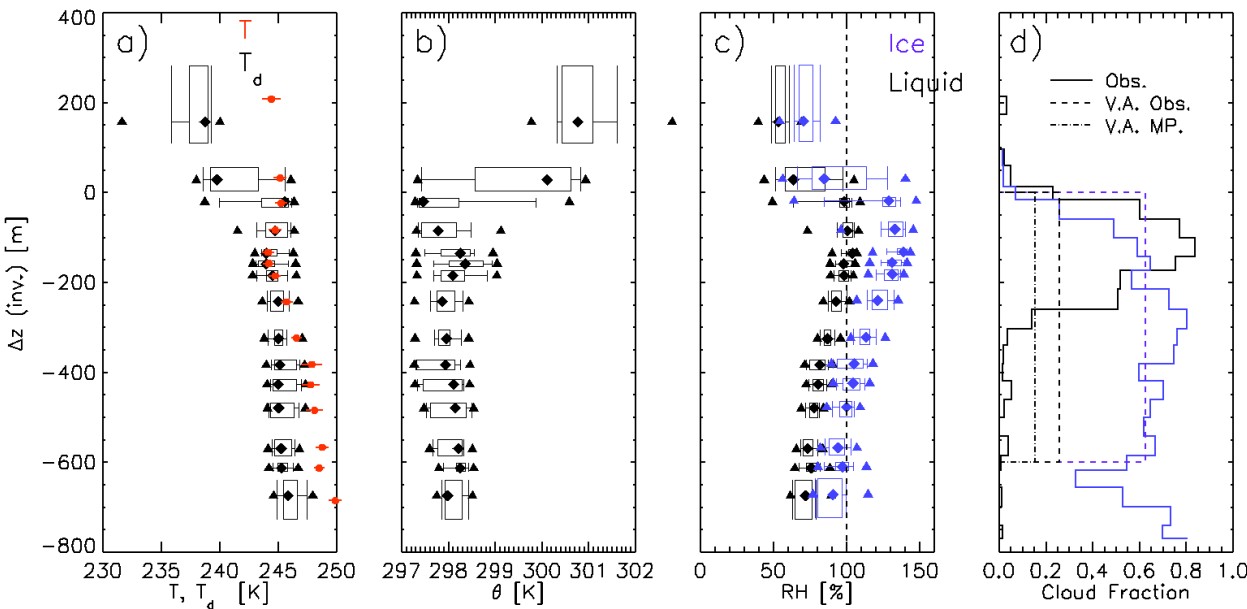

**Figure 4.** Vertical profiles of statistics w.r.t. inversion altitude, $\Delta z_{(inv.)}$, for (a) dew point (black), and mean and standard deviation of temperature (red), (b) potential temperature and (c) Relative Humidity w.r.t. Ice (blue) and Liquid (black). Boxes indicate the interquartile range (two-dimensional); bar-and-whiskers: $5^{th}$ and $95^{th}$ and filled triangles, $1^{st}$ and $99^{th}$ percentile. (d) Areal cloud fraction calculated from number concentration of liquid (CDP) (black solid) and ice particles (CIP15) (blue solid) over 40 m altitude bins with dashed lines showing the volume mean equivalent fraction over a 600 m deep layer (black - liquid, blue - ice), and calculated mixed-phase cloud fraction in this layer assuming maximum overlap (black dot-dashed line).

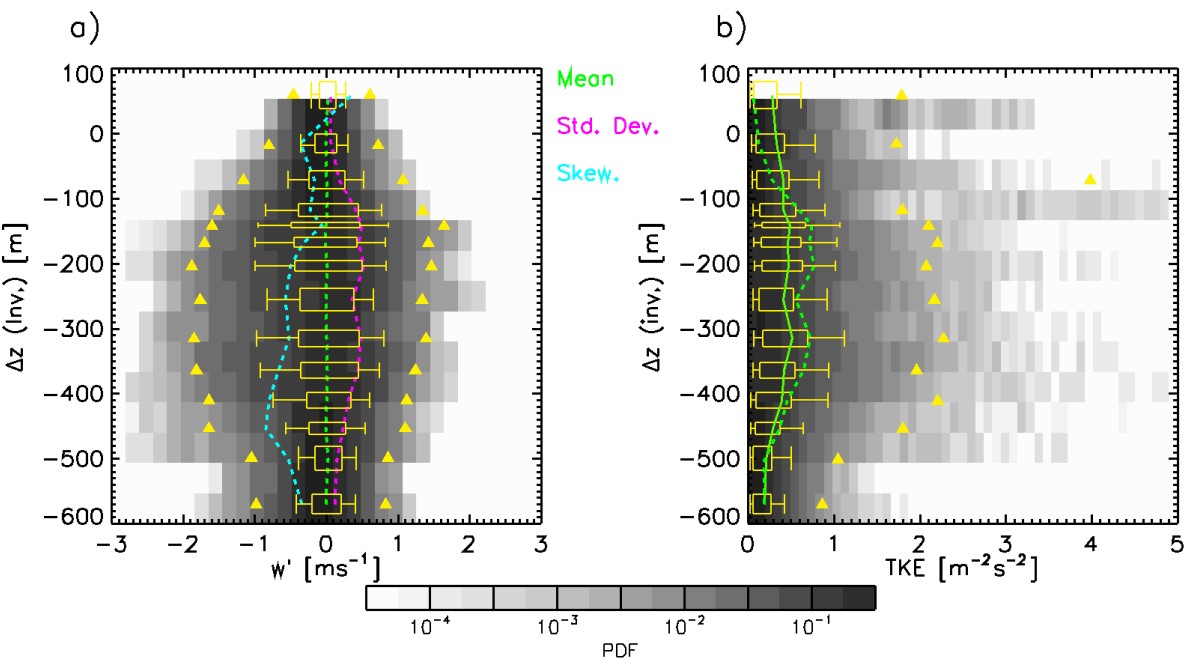

**Figure 5.** 2 dimensional histograms for 9 km filtered data, against distance from inversion altitude, for (a) vertical velocity fluctuations, w′ and (b) TKE, with PDF (Probability Distribution Function) of 32 Hz data (grey-scale - legend in lower panel) and percentiles (yellow). (a) Mean value of w′ (green) and standard deviation of w′ (magenta) and Skewness of w′ (cyan). (b) Mean value of TKE from three component winds (solid green) and with only the vertical component (see text for details) (dashed green).

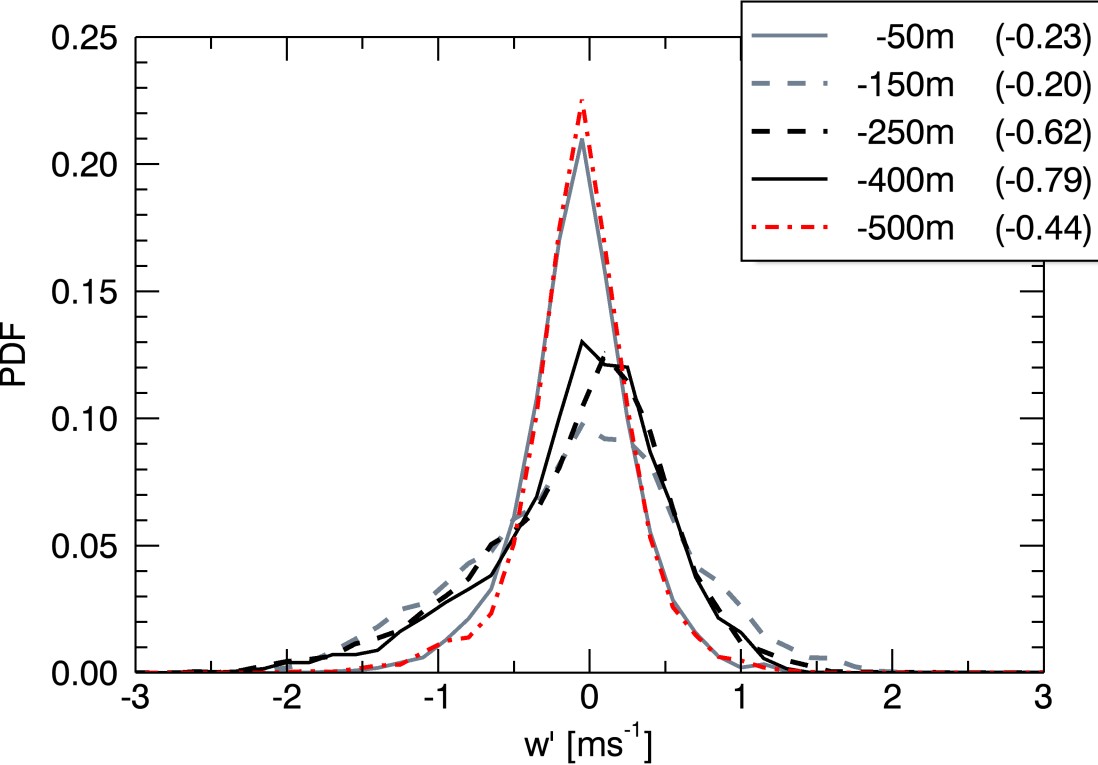

**Figure 6.** Vertical velocity fluctuations PDFs from 5 altitudes below inversion altitude with levels given in the legend. First value gives the centre altitude of the level, with the second number giving skewness of the vertical velocity distribution. Standard deviations of depth in each level were between 15 m and 50 m.

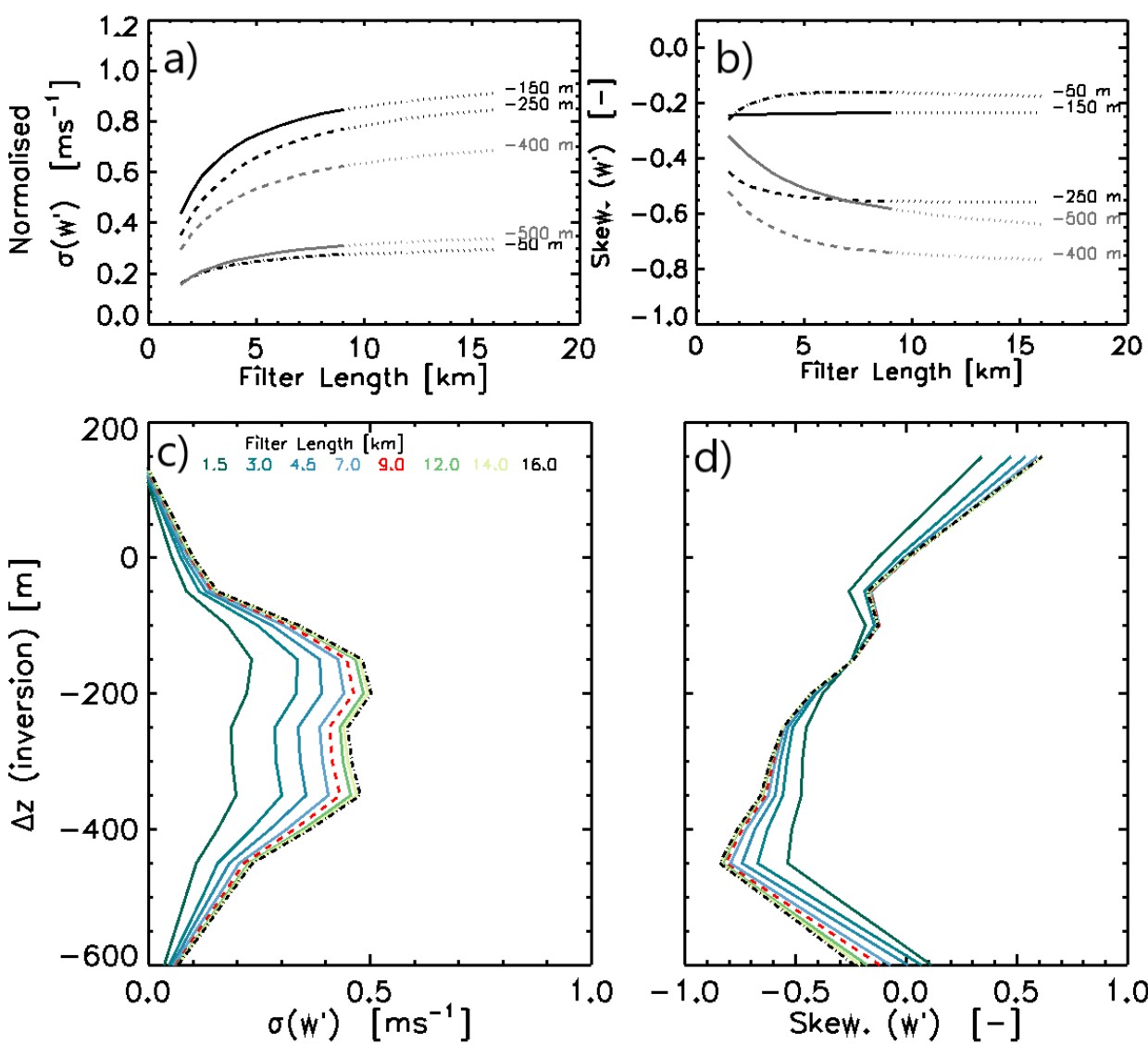

**Figure 7.** (a) Standard deviation of $w'$ as a function of filter length, normalised to the layer maximum value for 5 altitudes below the inversion altitude. Length scales longer than the chosen 9 km filter length are shown with dotted line to indicate how much of the variance is at scale that pass through this filter. (b) Skewness of distribution of $w'$ as a function of filter length, normalised to layer absolute maximum value, as (a). (c) Vertical profile of standard deviation of $w'$ relative to the inversion altitude, for a range of filter lengths from 1.5 km (blue) to 14 km (green) and the limit of 16 km with black-dot-dashed. 9 km is highlighted as red-dotted. (d) Vertical profile of skewness of the distribution of $w'$ relative to inversion altitude, as (c).

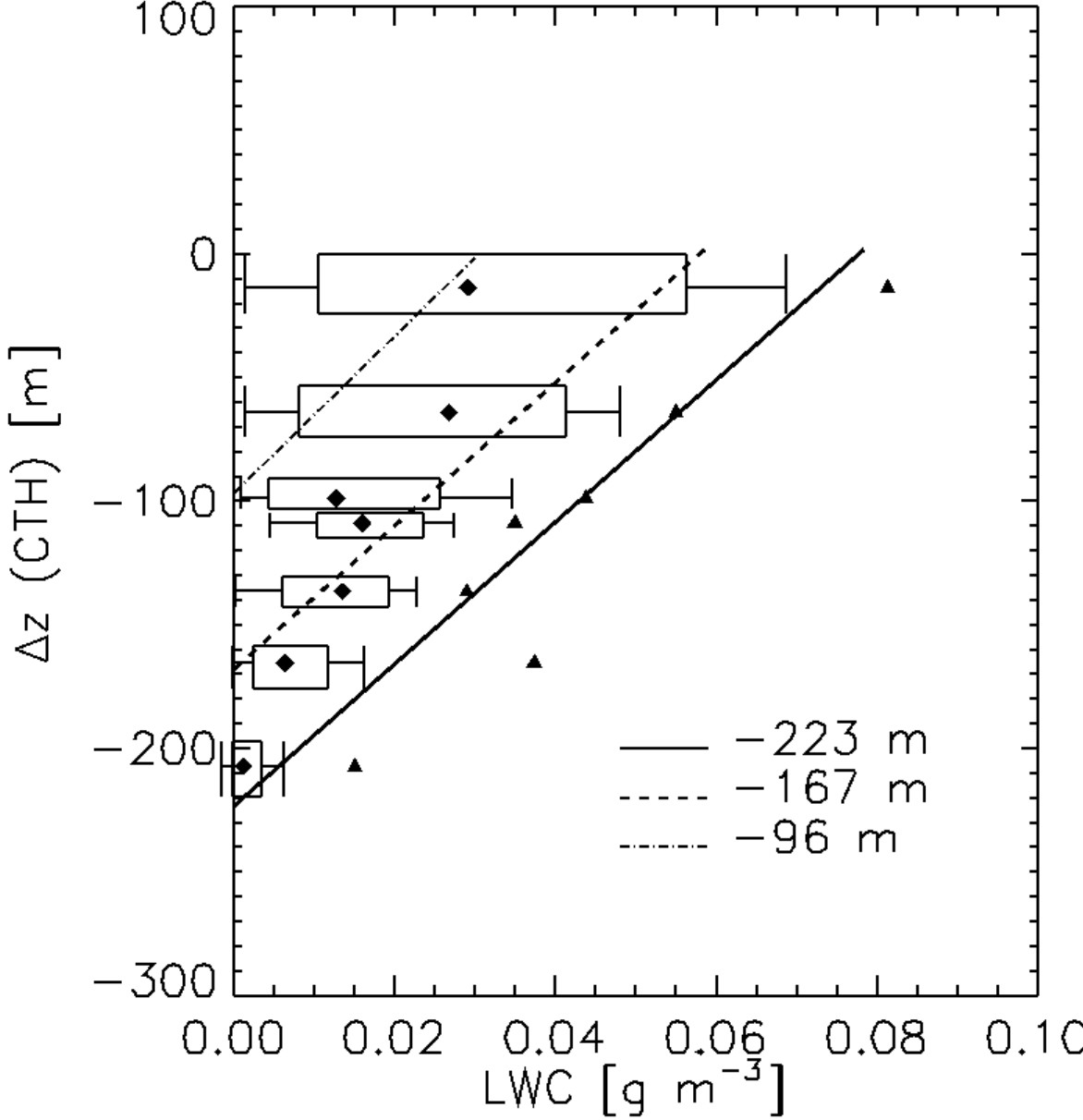

**Figure 8.** LWC profile percentiles relative to CTH for the mixed-phase cloud layer with data from Nevzorov LWC sensor. Theoretical adiabatic ascents are shown, starting from three altitudes (see legend for details) which correspond to potential cloud bases (see text for details). Statistics as Fig. 4.

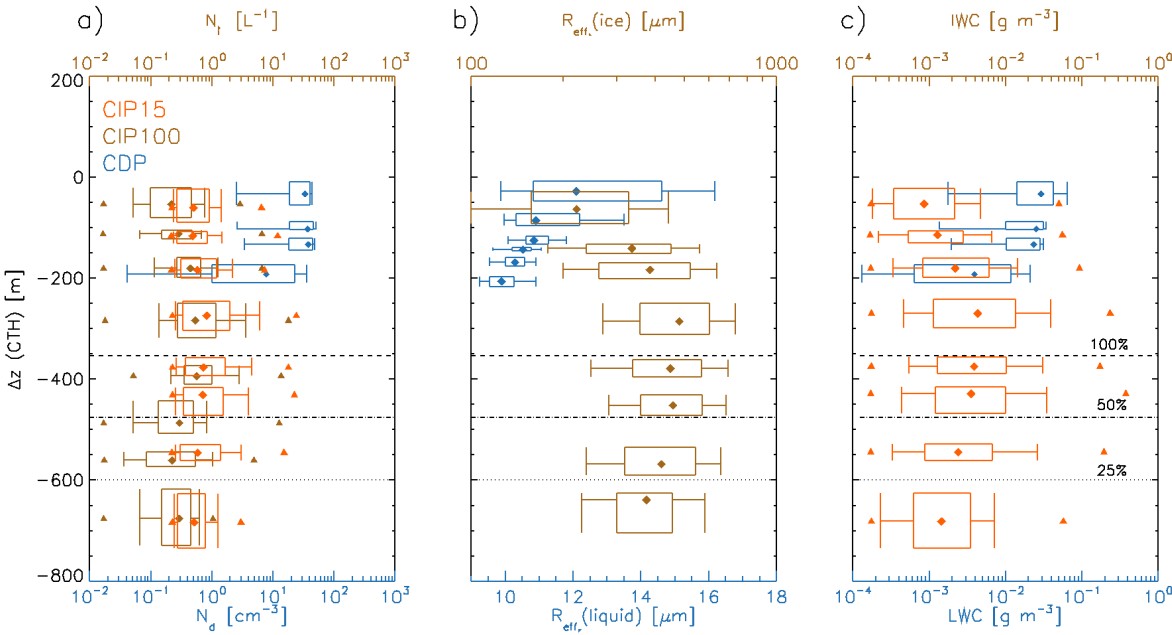

**Figure 9.** Vertical profiles relative to cloud top of (a) cloud particle number concentrations of liquid, $N_d$, (CDP [cm$^{-3}$]) (blue) and ice, $N_i$, (CIP15 [L$^{-1}$]) (orange) and (CIP100 [L$^{-1}$]) (light brown), (b) effective radius for liquid cloud particles, $R_{eff}$(liquid), (CDP) (blue) and ice particles $R_{eff}$(ice), (CIP100) (light brown - upper axis) (c) Condensed water content, CWC, with LWC (CDP) (blue) and IWC (CIP15) (orange). The altitude of observed frequency (areal fraction) of ice-supersaturation is shown for 100 % (black dashed line), 50 % (black dash-dotted line) and 25 % (black dotted line).

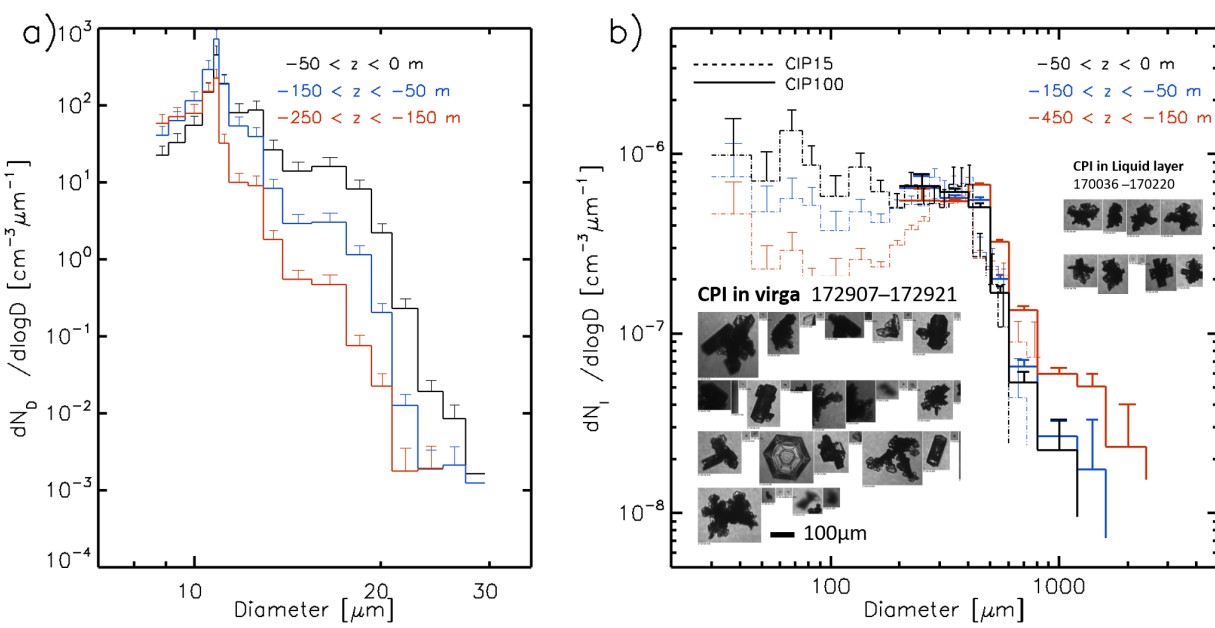

**Figure 10.** (a) Mean liquid cloud particle size distributions for three vertical levels as Table 3, with cloud top level (black), mid-cloud (blue) and cloud bases (red). Error bars represent standard deviations. (b) Ice particle size distribution from CIP15 (dashed line) and CIP100 (solid line) with colours as (a) and vertical levels as Table 3 with: -50 m < z < 0 m (black), -150 m < z < -50 m (blue), -250 m (liquid (a)), -450 m (ice (b)) < z < -150 m (red). Example particle images are taken from CPI probe in ice virga and from within the mixed phase cloud regions, and scale bar is shown for 100 $\mu$m length along with the time range of the imagery.

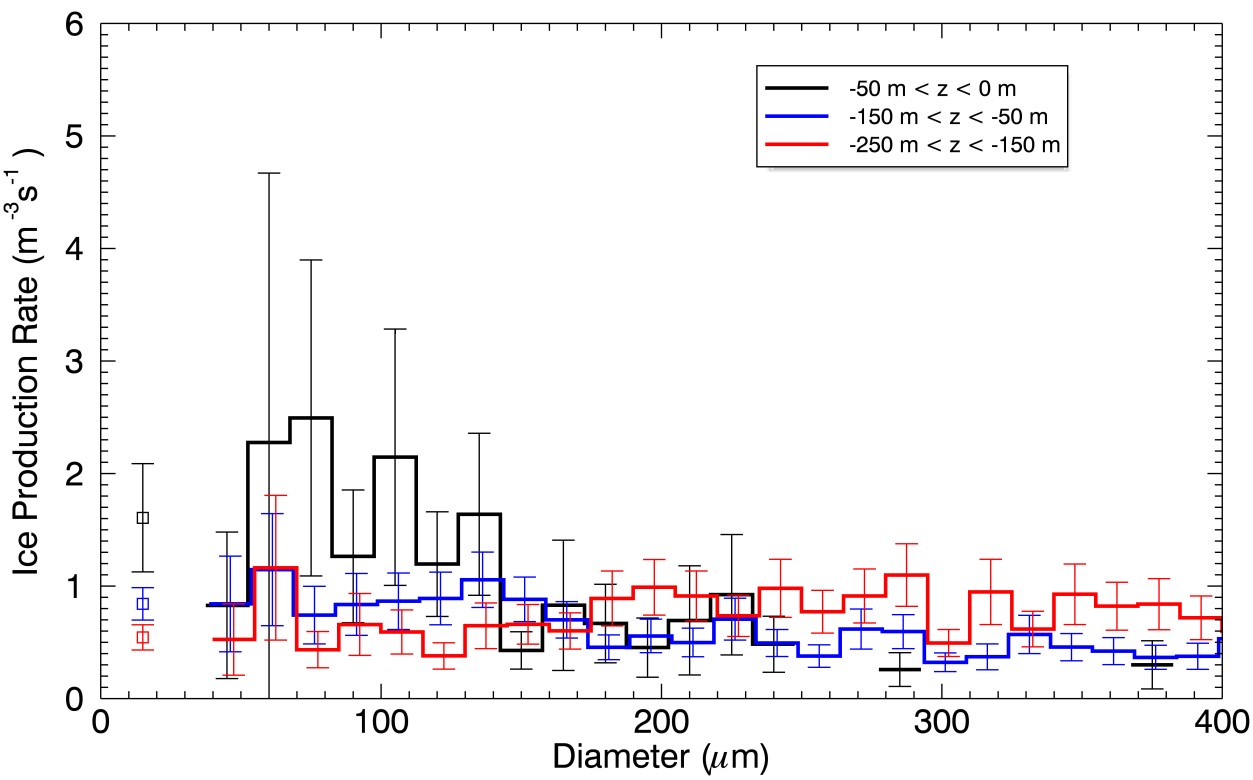

**Figure 11.** Ice production rate for three levels (Table 3) in altocumulus cloud with: -50 m < z < 0 m (black), -150 m < z < -50 m (blue), -250 m < z < -150 m (red), computed between adjacent channels of CIP15 data between 45 $\mu$m and 400 $\mu$m. Weighted mean values and errors are shown for the smallest particle size range (as Table 3) at all three levels with squares and errors bars.

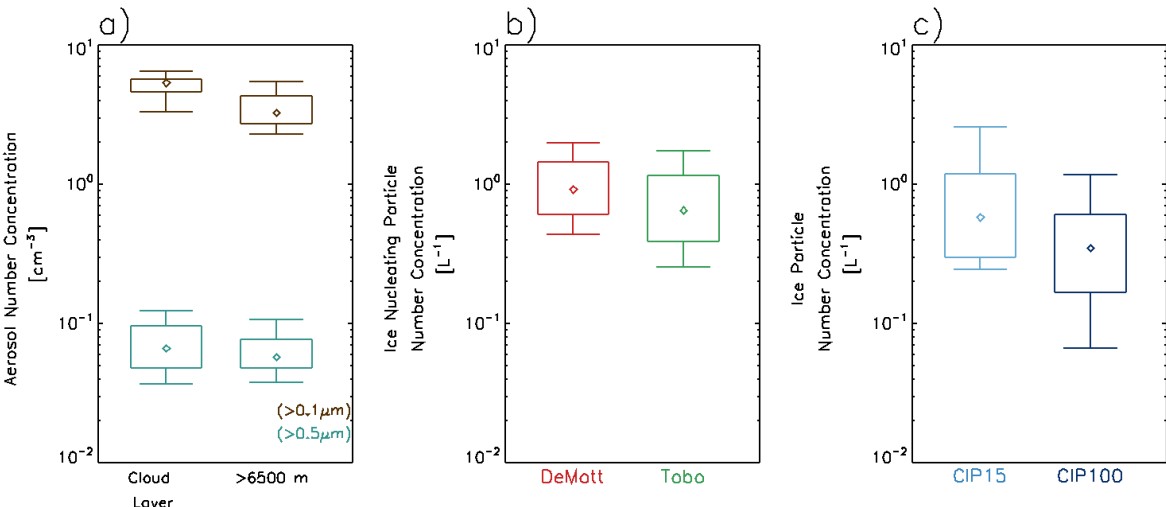

**Figure 12.** (a) Percentiles of aerosol particle number concentrations larger than 0.1 $\mu$m (brown) and 0.5 $\mu$m (teal) from PCASP for the cloud layer (left) and above 6500 m (right). (b) INP concentrations [L$^{-1}$] using DeMott et al. (2015) (red), and Tobo et al. (2013) (green). (c) Ice Particle number concentrations $N_i$ from CIP15 (light blue) and CIP100 (dark blue).

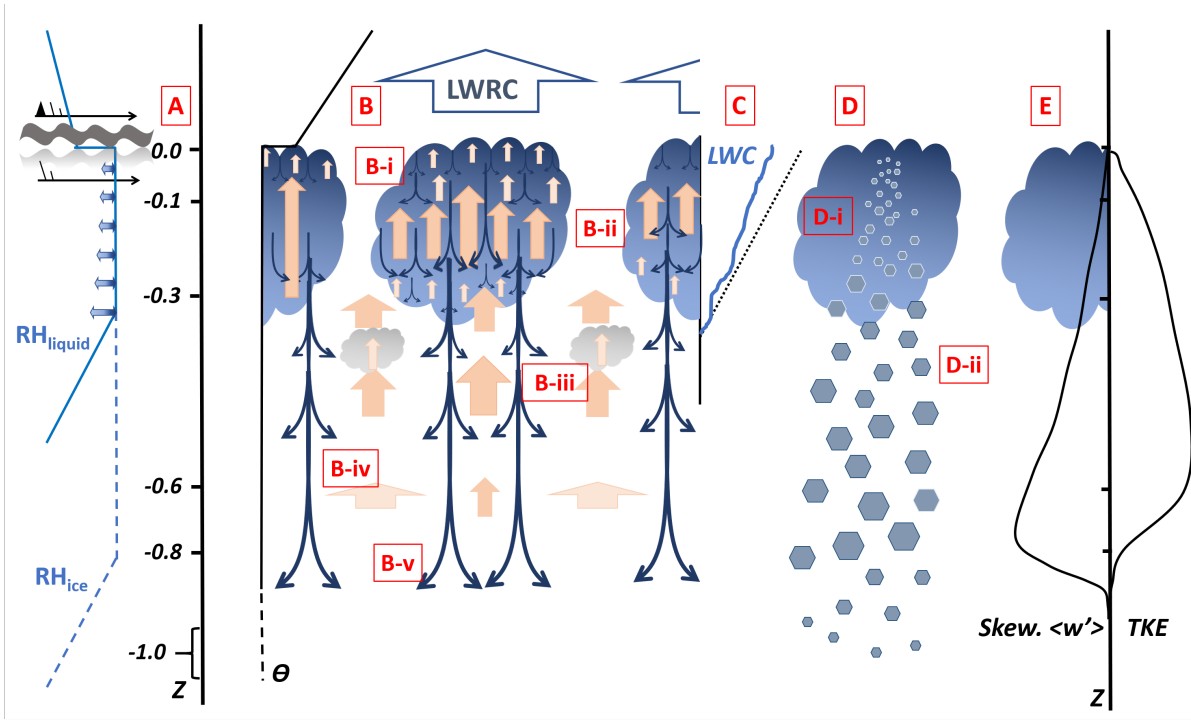

**Figure 13.** Schematic of processes that control altocumulus clouds as a function of normalised cloud layer depth below cloud top altitude, Z. The lower boundary is typically diffuse unlike for stratocumulus - which has the surface as a lower boundary (a) cloud formation through large scale ascent possibly augmented by wind shear or gravity wave activity acting at a stable interface in potential temperature, ($\theta$ - black trace) with a suitable relative humidity distribution (blue solid - RH w.r.t. liquid, and blue dashed - RH w.r.t. ice, and arrows indicate distribution up to 100 % )just below the temperature inversion. (b) Clouds (blue parcels) cool through Long-wave radiative cooling from cloud top, a process which imparts turbulence to the layer below with narrow strong downdraughts (blue arrows) and broad weaker updraughts (red arrows). (b-i) to (b-v) see text for details of the resulting turbulence structure (c) Turbulence acts to maintain the liquid water content profile (blue line) in a pseudo-adiabatic form (dashed black line). (d) At temperatures lower than 0° C, ice nucleation may occur, located within the liquid cloud, resulting in ice precipitating from the liquid cloud layer. (e) Turbulence Kinetic Energy peak occurs below the liquid cloud layer, with the skewness of the vertical velocity distribution occurring lower still.

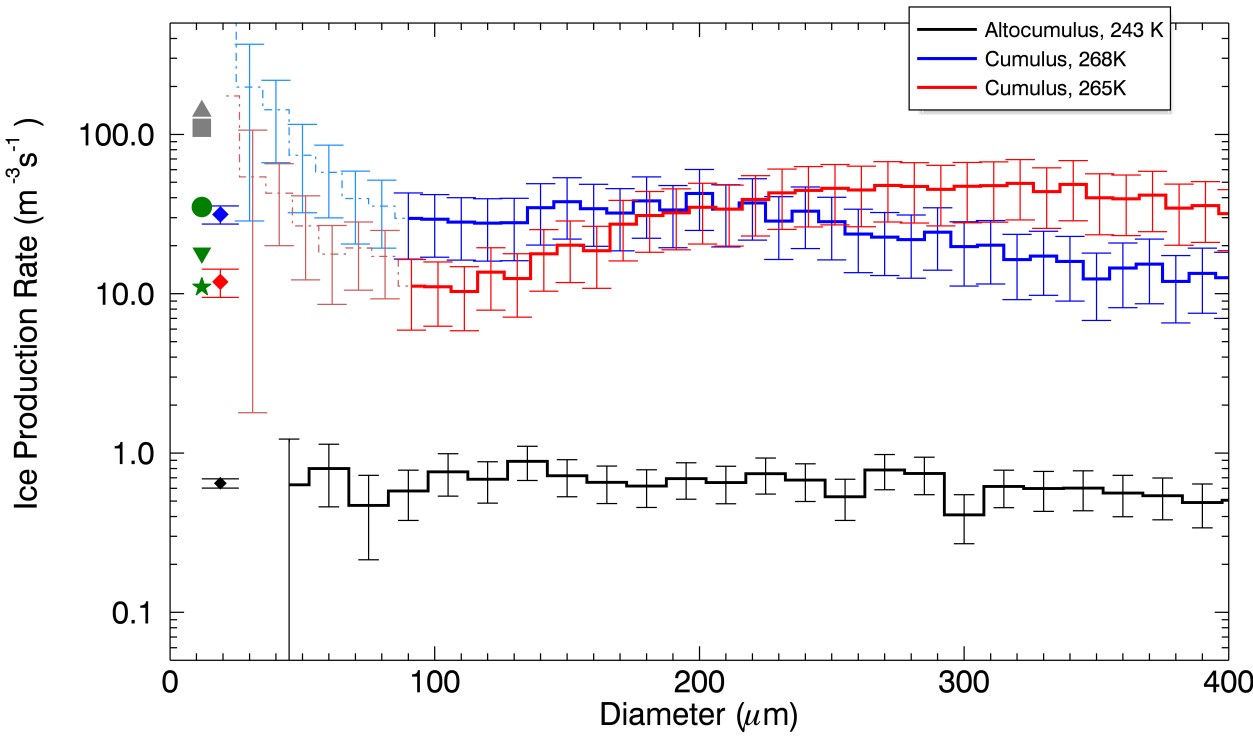

**Figure A1.** Ice production rate calculated between adjacent OAP bins smaller than 400 $\mu$m for the altocumulus clouds in this study at -30° C (black), and passes through two cumulus clouds in the HM temperature range, from flight B816 (Abel et al., 2017) at -5° C (blue) -8° C (red). Previous measurements in HM zone are shown from from HHC87 (green) and Taylor et al. (2016) (grey)

| Segment Type | Parameter | Units | Limit |
|---|---|---|---|
| SLR / Pro | $\frac{d}{dt}$ Heading | $^\circ\,\text{s}^{-1}$ | $\pm\,1.0$ |
| SLR / Pro | Roll | $^\circ$ | $\pm\,3.0$ |
| SLR | $\frac{d}{dt}$ Altitude | $\text{m s}^{-1}$ | $\leq \pm\,0.5$ |
| Pro | $\frac{d}{dt}$ Altitude | $\text{m s}^{-1}$ | $\pm\,5.0$ |

**Table 1.** Parameters and limits used in determining when aircraft is in a Straight and Level Run (SLR) and a slant Profile (Pro)

| Parameter | Mean [m] | Std. Dev. [m] | Range [m] |
|---|---|---|---|
| Cloud Top | 5656 | 150 | 432 |
| Inv. Top | 5666 | 146 | 439 |
| Lidar CTH | 5553 | 58 | 267 |
| Lidar CTH (30km) | <1 | 25 | 137 |
| Lidar CTH (3km) | <1 | 12 | 91 |
| Inv. Top Diff. | 30 | 25 | 112 |

**Table 2.** Cloud top height derived from CDP LWC, and inversion altitude derived from temperature profiles, and Lidar derived CTH (and with 30 km and 3 km trends removed) and difference (residual) between the measured inversion altitude and the derived inversion altitude: mean, standard deviation and range.

| | Altitude [m] | | | Sample Size [km] [km] [s] | | | $N_D > D$ [m$^{-3}$] where D > [$\mu$m] | | | Ice Production Rate [m$^{-3}$ s$^{-1}$] for D range [$\mu$m] | | |
|---|---|---|---|---|---|---|---|---|---|---|---|---|
| Name | Base | Top | T [K] | CDP | CIP | MP | 18.8 | 19.8 | 21.9 | 60–105 | 120–225 | 240–345 |
| Upper | -50 | 0 | $243.8 \pm 0.7$ | 44 | 29 | 31 | 84456 | 10944 | 1073 | $1.60 \pm 0.48$ | $0.59 \pm 0.11$ | $0.32 \pm 0.12$ |
| Intermediate | -150 | -50 | $243.8 \pm 0.6$ | 114 | 106 | 358 | 7411 | 687 | 152 | $0.84 \pm 0.14$ | $0.62 \pm 0.06$ | $0.43 \pm 0.04$ |
| Lower | -250 (-450*) | -150 | $244.2 \pm 0.5$ | 58 | 205 | 244 | 896 | 149 | 74 | $0.55 \pm 0.11$ | $0.63 \pm 0.06$ | $0.72 \pm 0.08$ |

**Table 3.** Vertical level (w.r.t. CTH) ranges for CDP PSD and CIP PSD and layer temperature ranges, with the distance spent sampling at that level along with time spent sampling mixed-phase (MP) clouds, number of cloud drops larger than given diameter, and size resolved ice production rates. *Lower level is deeper for CIP data.

| Filter Length [km] | Frequency [Hz] |
| --- | --- |
| 1.5 | 0.0933 |
| 2.0 | 0.0700 |
| 2.5 | 0.0560 |
| 3.0 | 0.0467 |
| 3.5 | 0.0400 |
| 4.0 | 0.0350 |
| 4.5 | 0.3111 |
| 5.0 | 0.0280 |
| 6.0 | 0.0234 |
| 7.0 | 0.2000 |
| 8.0 | 0.0175 |
| 9.0 | 0.0156 |
| 12.0 | 0.0117 |
| 14.0 | 0.0100 |
| 16.0 | 0.0088 |

**Table A1.** Butterworth Filter Lengths and Frequencies, used to investigate turbulence.