# Peer review of "The Structure of Turbulence and mixed-phase Cloud Microphysics in a Highly Supercooled Altocumulus Cloud"

_Atmospheric Chemistry and Physics, 2019_

## Referee Comment (RC1) · Anonymous Referee #1 · 16 Oct 2019

Review of 'The structure of turbulence and mixed-phase cloud microphysics in a highly supercooled altocumulus cloud" Barrett et al

The authors present a case study of a mixed-phase altocumulus cloud observed over the N. Atlantic. In situ observations collected from an instrumented aircraft provide details of the cloud thermodynamic, microphysical, and turbulent structure. The roughly 600 m thick cloud consisted of a relatively thin liquid top with mixed phase below, giving way to all ice in the lowest ∼half of the cloud. Observations were obtained through a combination of straight/level legs and sawtooth legs over ∼10 km horizontal distance. The authors carefully designed a coordinate system referencing the inversion

that topped the cloud layer, and by letting the cloud advect in time in relation to this coordinate system, reference all measurements to a single vertical coordinate.

The manuscript is well written and presents a comprehensive set of observations from a single case. The authors' analyses are thorough and their conceptual model presented in the discussion is well supported by the measurements. The authors also do a good job of placing their observations in the context of previous work. I recommend publishing after they address a few minor comments.

Minor Comments I don't believe the authors make a strong connection between the increasing negative skewness of w' with increasing distance beneath cloud top and what they assert is driving this, specifically longwave radiative cooling at cloud top. This assertion is stated both in the abstract and in the conclusions and while I agree their conceptual model fits well with that presented in other papers, such as Schmidt et al (2014) for instance, they really don't provide any evidence or even an explanation why radiative cooling should result in w' profiles as those observed.

Last sentence in the abstract: "...used to evaluate numerical simulations of altocumulus clouds at all scales from eddy resolving to climate." The authors are mixing up spatial scales (eddy-resolving) with temporal (climate). I understand the point they are trying to make, but this should be reworded in my opinion.

Line 93: I'm pretty sure the thermometer did not respond at 32 Hz, even though measurements may have been reported at this high rate. Authors should provide actual response rate of the instrument. Also, many of the measurements were conducted in cloud. Were corrections made for wetting of the element? Or how were in-cloud (liquid) temperatures determined?

Page 4 and variety of locations in manuscript regarding discussion of using the CIP15 (and CIP100) to determine NI (ice crystal concentrations). Some number of 30-60 (or maybe even 90) micron particles measured by the CIP15 are likely liquid droplets. First, some liquid in these larger size ranges might exist and the CIP15 does not provide sufficient resolution to distinguish between liquid and ice for such small particles. Second, smaller, out-of-focus droplets will appear larger. This could lead to an over-estimation of the ice crystal concentration nearer cloud top where liquid droplets are largest. How do the results change if the analysis were to ignore all measurements from the CIP15 with diameters smaller than 90 microns?

Bottom of page 6, line 187: the impact of wind-shear across the inversion and resultant contamination of the filter should only occur near the inversion, correct? If that is the case, then why substitute w' in place of u' and v' throughout the depth of the profiles. I think you should only do that near cloud top.

Around Line 280, discussing estimate of INC based on aerosol larger than 0.5 um. The authors provide an estimate based on DeMott et al (2015) and then appear to 'adjust' (increase) that based on a correction factor of 3. But DeMott et al's equation (2) includes CF (calibration factor as they call it). Thus to be consistent, authors should only report the larger value; the lower value of 0.3 L-1 doesn't make any sense, or at least is NOT CONSISTENT with DeMott et al (2015).

Technical Comments Line 4: change 'Turbulence data' to 'Turbulence measurements'

Line 16: '. . .mixed-phase conditions began within a few metres. . .' began is probably not the right work, it implies time. I sugges something like '. . .mixed phase conditions were observed within a few metres of cloud top extending downward through the cloud'

Line 89: abbreviation of 'knots' should it be 'kts' ??

Line 275: 'Fig. 12@(b)' ??
* * *

---

## Referee Comment (RC2) · Anonymous Referee #2 · 6 Nov 2019

The authors describe an altocumulus cloud that was observed by aircraft. Turbulence and microphysical properties are recorded. Conjectures about the responsible physical processes are made. Only one cloud was observed, but altocumulus clouds are transient, and hence aircraft measurements are rare. So every new observed cloud case is useful. The observations of negative vertical velocity skewness below the liquid cloud base are interesting.

Minor comments:

Abstract: "The turbulence spectrum is observed to have an increasingly negative skewness with distance below cloud top, confirming that longwave radiative cooling from the

liquid layer cloud is the source of turbulence kinetic energy." Personally, I would choose the more cautious term "suggesting" in lieu of the bolder term "confirming". I have no doubt that cloud-top radiative cooling is relevant, but without a complete budget of TKE, it is hard to know the relative magnitude of each source.

Lines 40–41: "The GCM simulations are found to have too little cloud in the mid-levels, resulting in a warm bias in sea surface temperatures," The authors might also want to mention Hartmann et al. (1992, J. Climate, see Fig. 22 and Table 1), which suggests that thin, mid-level clouds have little net radiative impact at the top of the atmosphere.

Lines 74–75: "which was predominantly from the south and ranged in strength from 6 m s −1 at the southern end of the flight track to 8 m s −1 in the north." It might also be worth recording the mean vertical wind shear within and near the cloud, because wind shear is associated with generation of turbulence.

Figure 8: On this figure or in its caption, please clarify what the diamond symbols mean, and how the reader is to infer whether the profiles are adiabatic by comparing to the many lines drawn on the figure. The description in Lines 218–224 is also a little unclear to me. E.g., a theoretical adiabatic ascent line is drawn starting at 136 m below cloud top. Was there clear, ascending air observed at this altitude? If not, what does the adiabatic line represent?

Fig. 10b: Are the habits of the ice particles photographed by CPI so complex because of aggregation or instead polycrystal formation because of defects (grain boundaries) in the crystals? The text in Lines 256–259 seems to suggest both. Is there a way to distinguish the two?

Fig. 13: "cloud formation through wind shear or gravity wave activity acting at a stable interface in potential temperature" Is it possible that the cloud forms from mesoscale or synoptic-scale lifting rather than gravity waves, and that the cloud-top inversion forms later through cloud-top radiative cooling? Is any estimate of large-scale vertical velocity available?

---

## Author Comment (AC1) · 5 Dec 2019

We thank the reviewer for their helpful comments and suggestions.

**1   Minor Comments**

*I dont believe the authors make a strong connection between the increasing negative skewness of w with increasing distance beneath cloud top and what they assert is driving this, specifically longwave radiative cooling at cloud top. This assertion is stated*

[Figure]

*both in the abstract and in the conclusions and while I agree their conceptual model fits well with that presented in other papers, such as Schmidt et al (2014) for instance, they really dont provide any evidence or even an explanation why radiative cooling should result in w profiles as those observed.*

Long-wave Radiative Cooling and skewness. We have added comment in the discussion relating to the Hogan 2009 study which observed the skewness profile of nocturnal stratocumulus to be similar to the profile we observed. This is the basis for our assertion that the source of the turbulence is at cloud top.

*Last sentence in the abstract: ...used to evaluate numerical simulations of altocumulus clouds at all scales from eddy resolving to climate. The authors are mixing up spatial scales (eddy-resolving) with temporal (climate). I understand the point they are trying to make, but this should be reworded in my opinion.*

Abstract text clarified to account for spatial and temporal scale confusion.

*Line 93: Im pretty sure the thermometer did not respond at 32 Hz, even though measurements may have been reported at this high rate. Authors should provide actual response rate of the instrument. Also, many of the measurements were conducted in cloud. Were corrections made for wetting of the element? Or how were in-cloud (liquid) temperatures determined?*

Some of the temperature measurements were in fact collected in cloud. We expect that the low liquid water contents (less than 0.05 g m$^{-3}$) in the cloud result in a negligible impact on the cooling of the non-de-iced sensor housing. A review by Sinkevich and Lawson (2005) of temperature measurements in convective clouds would seem to confirm this. Figure 4 of that paper shows the difference a radiometric measurement of temperature and an immersion sensor as a function of liquid water content. For the liquid water contents that we observed the temperature difference or 0.05 °C is an order of magnitude lower than the quoted uncertainty of our temperature sensors.

The reviewer notes that the temperature sensor is unlikely to respond as fast as 32 Hz even though it is recorded at the rate. We agree and in-fact here we are reporting data at 1 Hz and have amended the text to reflect this.

We do however use the turbulent wind data at recorded at 32 Hz. We computed power spectra from one of the geometrically level flight legs and found that all three wind components follow a -5/3 power law out to at least 10 Hz, possibly closer to the Nyquist frequency of 16 Hz. We reprocessed the wind components after applying a smoothing window (to give data at a frequency of less than 10 Hz) and found that the TKE was not impacted by more than 5 percent, with skewness values not impacted.

*Page 4 and variety of locations in manuscript regarding discussion of using the CIP15 (and CIP100) to determine NI (ice crystal concentrations). Some number of 30-60 (or maybe even 90) micron particles measured by the CIP15 are likely liquid droplets. First, some liquid in these larger size ranges might exist and the CIP15 does not provide sufficient resolution to distinguish between liquid and ice for such small particles. Second, smaller, out-of-focus droplets will appear larger. This could lead to an over-estimation of the ice crystal concentration nearer cloud top where liquid droplets are largest. How do the results change if the analysis were to*

We refer to the CIP15 (which responds to particles larger than 15 microns) number concentrations for ice in multiple places throughout the text. Whilst we agree that some artefacts may be present in the CIP15 ice data at sizes between 60 and 90 microns, when we only consider the CIP15 data for particles larger than 90 microns we find only minimal difference in the statistics of the vertical profiles of ice number concentrations, that do not alter the conclusions. As part of the quality control process we looked at the histograms of number concentrations from CIP15 and CIP100 (which responds to particles larger than 100 microns) and considered individual channels concentrations from CIP15. The figure here shows the distribution of particle concentrations from CIP15 (green) and CIP100 (orange). Data from CIP15 are clearly bimodal, with the higher concentration mode having number concentrations greater than 0.5 L$^{-1}$. Plotting only

the data from the smallest size channel (Channel 0) on CIP15 (dark red) shows that these particles are the ones smaller than 15 microns, and are almost certainly part of the liquid cloud population. CIP100 does not observed this mode. Channel 1 only from CIP15 (30 micron particles) (red) and channel 2 (45 micron particles) (pink) do not observe this mode. The total concentration of particles larger than a particular size is shown in the blue colours, with particles larger than 30, 45, 75, 135 and 255 microns shown in ever lighter shades of blue. These distributions shows only the ice population and look similar to the observations from CIP100. We took this to show that in this situation with the observed liquid cloud population on the day that the CIP15 only responded to the liquid cloud in the smallest size channel, bin 0 (less than 15 microns). We therefore have confidence that bins larger than this are responding only to ice, but leave a buffer of two adjacent bins.

Any contamination by small particles would not impact the IWC calculation significantly. We do not use the CIP 15 for effective radius calculations, instead we rely on CIP100.

*Bottom of page 6, line 187: the impact of wind-shear across the inversion and resultant contamination of the filter should only occur near the inversion, correct? If that is the case, then why substitute w in place of u and v throughout the depth of the profiles. I think you should only do that near cloud top.*

In figure 5 we have added a trace showing the TKE estimate when considering all three wind components. This shows the good performance of just considering w′ when above -100 m, and that the estimate only using w′ is biased high compared to the three wind component estimate, when below -100 m. This is discussed in the text.

*Around Line 280, discussing estimate of INC based on aerosol larger than 0.5 um. The authors provide an estimate based on DeMott et al (2015) and then appear to adjust (increase) that based on a correction factor of 3. But DeMott als equation (2) includes CF (calibration factor as they call it). Thus to be consistent, authors should only report the larger value; the lower value of 0.3 L-1 doesnt make any sense, or at*

*least is NOT CONSISTENT with DeMott et al (2015).*

We agree with the reviewer. Reference to uncorrected INP concentration is removed from the figure and the text.

**2 Technical Comments**

*Line 4: change Turbulence data to Turbulence measurements*

We have changed Turbulence data to Turbulence measurements

*Line 16: . . .mixed-phase conditions began within a few metres. . .began is probably not the right work, it implies time. I sugges something like . . .mixed phase conditions were observed within a few metres of cloud top extending downward through the cloud*

We agree, began was not the correct word. Text has been altered.

*Line 89: abbreviation of knots should it be kts ??*

Abbreviation changed to kts

*Line 275: Fig. 12@(b) ?*

typo corrected

A.A. Sinkevich and R.P. Lawson, A Survey of Temperature Measurements in Convective Clouds, J. Appl. Met. 44, 1133-1145, 2005
* * *
Histogram Density

CIP15
CIP100

CIP15>30$\mu m$
CIP15>45$\mu m$
CIP15>75$\mu m$
CIP15>135$\mu m$
CIP15>255$\mu m$

CIP15 ch0=15$\mu m$
CIP15 ch1=30$\mu m$
CIP15 ch2=45$\mu m$

Number Concentration [L$^{-1}$]

**Fig. 1.** Cloud particle number concentration histograms from CIP15 and CIP100

---

## Author Comment (AC2) · 5 Dec 2019

We thank the reviewer for the helpful comments and positive response to the paper.

**1   Minor comments:**

*Abstract: The turbulence spectrum is observed to have an increasingly negative skewness with distance below cloud top, confirming that longwave radiative cooling from the liquid layer cloud is the source of turbulence kinetic energy.  Personally, I would*

[Figure]

*choose the more cautious term suggesting in lieu of the bolder term confirming. I have
no doubt that cloud-top radiative cooling is relevant, but without a complete budget of
TKE, it is hard to know the relative magnitude of each source.*

We agree that the data do not confirm turbulence generated through longwave radiative
cooling and agree that "suggests" is more appropriate word for the abstract.

*Lines 40-41: The GCM simulations are found to have too little cloud in the mid-levels,
resulting in a warm bias in sea surface temperatures, The authors might also want to
mention Hartmann et al. (1992, J. Climate, see Fig. 22 and Table 1), which suggests
that thin, mid-level clouds have little net radiative impact at the top of the atmosphere.*

The clouds in the study would indeed be of Type 3 (Hartmann 1992) as they are indeed
optically thin, under this definition. We add a caveat and reference to the paper to state
that while regional the clouds may have some radiative impact, the global significance
of the cloud type is low.

*Lines 74,75: which was predominantly from the south and ranged in strength from 6 m
s-1 at the southern end of the flight track to 8 m s-1 in the north. It might also be worth
recording the mean vertical wind shear within and near the cloud, because wind shear
is associated with generation of turbulence.*

We add comment that mentions the potential for turbulence generated through shear
in these clouds.

*Figure 8: On this figure or in its caption, please clarify what the diamond symbols
mean, and how the reader is to infer whether the profiles are adiabatic by comparing
to the many lines drawn on the figure. The description in Lines 218-224 is also a little
unclear to me. E.g., a theoretical adiabatic ascent line is drawn starting at 136 m below
cloud top. Was there clear, ascending air observed at this altitude? If not, what does
the adiabatic line represent?*

Caption refers to Figure 6 for details of symbols. There are observations of updraughts

in unsaturated air through the depth of the cloud system to as low as 500 m below the liquid cloud top. Cloud base was visually observed to be non-uniform, in the turbulence mixing- (but therefore not well-mixed) layer. This is supported by the Schmidt observations. The adiabatic ascents are shown to demonstrate that the in-cloud LWC distribution could arise as a result of this variable cloud base - which is in turn driven by the cloud top cooling generated turbulence, rather than by dilution through entrainment either at cloud top or laterally. The figure has been replotted to only show 3 ascents for clarity.

*Fig. 10b: Are the habits of the ice particles photographed by CPI so complex because of aggregation or instead polycrystal formation because of defects (grain boundaries) in the crystals? The text in Lines 256,–259 seems to suggest both. Is there a way to distinguish the two?*

We have not been able to disentangle the effects of polycrystaline growth from the impact of ice particle aggregation in the CPI imagery. Shadow imaging probe data are also not able to distinguish between the two. It is likely that both processes are active in the cloud. Attempting to quantify the aggregation rate in the clouds based on particle size number concertation etc, was deemed to be too complex and uncertain and so beyond the scope of the work. It may impact the ice production rate estimate to some degree.

*Fig. 13: cloud formation through wind shear or gravity wave activity acting at a stable interface in potential temperature Is it possible that the cloud forms from mesoscale or synoptic-scale lifting rather than gravity waves, and that the cloud-top inversion forms later through cloud-top radiative cooling? Is any estimate of large-scale vertical velocity available?*

Following on from the discussion regarding turbulence generation from cloud top cooling in the abstract, we agree, that without a full turbulence budget it is not clear the source of the turbulence, but we also suspect that the wind shear has produced some

component of the TKE. There is no estimate of large scale ascent (unless we resort to a reanalysis product), but it is certain that the clouds were in the warm sector of a mid-latitude cyclone, and were experiencing slow large scale ascent along isentropic surfaces in that region. Mention of large scale ascent has been added to the Figure 13 caption, in line with elsewhere in the text. We suspect the whilst this would produce supersaturation at some stage, the likelihood is that the first liquid clouds were produced when additional uplift from either shear induced wave activity at the top of the warm sector, or gravity wave activity at the same surface resulting from convection in the frontal zones, was present. This cloud would then cool to space and thus begin to produce the turbulence that could maintain the liquid cloud layer, and lead on to ice production.

We agree that the cooling may produce the inversion later on, and have amended the text to say so.

---

## Author Comment (AC3) · 7 Jan 2020

We thank the reviewer for their helpful comments and suggestions.

The reviewer comments are in *italics*, author response in plain text, and modified manuscript text in **boldface**.

[Figure]

**1 Minor Comments**

*I dont believe the authors make a strong connection between the increasing negative skewness of w with increasing distance beneath cloud top and what they assert is driving this, specifically longwave radiative cooling at cloud top. This assertion is stated both in the abstract and in the conclusions and while I agree their conceptual model fits well with that presented in other papers, such as Schmidt et al (2014) for instance, they really dont provide any evidence or even an explanation why radiative cooling should result in w profiles as those observed.*

Long-wave Radiative Cooling and skewness. We have added comment in the discussion relating to the Hogan 2009 study which observed the skewness profile of nocturnal stratocumulus to be similar to the profile we observed. This is the basis for our assertion that the source of the turbulence is at cloud top.

**The negatively skewed vertical velocity distribution (Fig. 6) is similar to that found in nocturnal (Nicholls, 1989). Hogan et al. (2009) (Fig. 13 (b)) found a similar stratocumulus (Nicholls, 1989; Hogan et al., 2003a), with a profile of skewness for LWRC driven nocturnal stratocumulus clouds using ground based measurements, in contrast with the profiles of skewness that were obtained when surface heating driven cumulus clouds were overhead. The turbulence kinetic...**

*Last sentence in the abstract: ...used to evaluate numerical simulations of altocumulus clouds at all scales from eddy resolving to climate. The authors are mixing up spatial scales (eddy-resolving) with temporal (climate). I understand the point they are trying to make, but this should be reworded in my opinion.*

Abstract text clarified to account for spatial and temporal scale confusion.

**These high resolution in situ measurements support previous remotely-sensed observations from both ground based and space borne instruments, and could be used to evaluate numerical model simulations of altocumulus clouds at spa-**

**tial scales from eddy resolving to global numerical eather prediction models and climate simulations.**

*Line 93: Im pretty sure the thermometer did not respond at 32 Hz, even though measurements may have been reported at this high rate. Authors should provide actual response rate of the instrument. Also, many of the measurements were conducted in cloud. Were corrections made for wetting of the element? Or how were in-cloud (liquid) temperatures determined?*

Some of the temperature measurements were in fact collected in cloud. We expect that the low liquid water contents (less than 0.05 g m$^{-3}$) in the cloud result in a negligible impact on the cooling of the non-de-iced sensor housing. A review by Sinkevich and Lawson (2005) of temperature measurements in convective clouds would seem to confirm this. Figure 4 of that paper shows the difference a radiometric measurement of temperature and an immersion sensor as a function of liquid water content. For the liquid water contents that we observed the temperature difference or 0.05 °C is an order of magnitude lower than the quoted uncertainty of our temperature sensors.

**There was no evidence of contamination on this sensor housing due to the presence of condensed liquid water when compared against the deiced sensor.**

The reviewer notes that the temperature sensor is unlikely to respond as fast as 32 Hz even though it is recorded at the rate. We agree and in-fact here we are reporting data at 1 Hz and have amended the text to reflect this.

**Temperature measurements were provided recorded at 32 Hz by a non-de-iced Goodrich Type 102 platinum-resistance thermometer and reported at 1 Hz.**

We do however use the turbulent wind data at recorded at 32 Hz. We computed power spectra from one of the geometrically level flight legs and found that all three wind components follow a -5/3 power law out to at least 10 Hz, possibly closer to the Nyquist frequency of 16 Hz. We reprocessed the wind components after applying a smoothing

window (to give data at a frequency of less than 10 Hz) and found that the TKE was not impacted by more than 5 percent, with skewness values not impacted.

*Page 4 and variety of locations in manuscript regarding discussion of using the CIP15 (and CIP100) to determine NI (ice crystal concentrations). Some number of 30-60 (or maybe even 90) micron particles measured by the CIP15 are likely liquid droplets. First, some liquid in these larger size ranges might exist and the CIP15 does not provide sufficient resolution to distinguish between liquid and ice for such small particles. Second, smaller, out-of-focus droplets will appear larger. This could lead to an over-estimation of the ice crystal concentration nearer cloud top where liquid droplets are largest. How do the results change if the analysis were to*

We refer to the CIP15 number concentrations for ice in multiple places throughout the text. Whilst we agree that some artefacts may be present in the CIP15 ice data at sizes between 60 and 90 microns, when we only consider the CIP15 data for particles larger than 90 microns we find only minimal difference in the statistics of the vertical profiles of ice number concentrations, that do not alter the conclusions. As part of the quality control process we looked at the histograms on number concentrations from CIP15 and CIP100 and considered individual channels from CIP15. The figure below (Fig. 1) shows the distribution of particle concentrations from CIP15 (green) and CIP100 (orange). Data from CIP15 are clearly bimodal, with the higher concentration mode having number concentrations greater than 0.5 $L^{-1}$. Plotting only the data from the smallest size channel (Channel 0) on CIP15 (dark red) shows that these particles are the ones smaller than 15 microns, and are almost certainly part of the liquid cloud population. CIP100 does not observed this mode. Channel 1 only from CIP15 (30 micron particles) (red) and channel 2 (45 micron particles) (pink) do not observe this mode. The total concentration of particles larger than a particular size is shown in the blue colours, with particles larger than 30, 45, 75, 135 and 255 microns shown in ever lighter shades of blue. These distributions shows only the ice population and look similar to the observations from CIP100. We took this to show that in this situation with

the observed liquid cloud population on the day that the CIP15 only responded to the liquid cloud in the smallest size channel, bin 0 (less than 15 microns). We therefore have confidence that bins larger than this are responding only to ice, but leave a buffer of two adjacent bins.

Any contamination by small particles would not impact the IWC calculation significantly. We do not use the CIP 15 for effective radius calculations, instead we rely on CIP100.

*Bottom of page 6, line 187: the impact of wind-shear across the inversion and resultant contamination of the filter should only occur near the inversion, correct? If that is the case, then why substitute w in place of u and v throughout the depth of the profiles. I think you should only do that near cloud top.*

In figure 5 we have added a trace showing the TKE estimate when considering all three wind components. This shows the good performance of just considering $w'$ when above -100 m, and that the estimate only using $w'$ is biased high compared to the three wind component estimate, when below -100 m. This is discussed in the text. Below this altitude the estimate of TKE using only $w'$ is biased high as compared to the full three dimensional estimate.

A correction to Figure 5 Panel (b), now shows the mean values trace as green, rather than the previous incorrect magenta.

The new Figure 5 caption reads: **Figure 5. 2 dimensional histograms for 9 km filtered data, against distance from inversion altitude, for (a) vertical velocity fluctuations, $w'$ and (b) TKE, with PDF (Probability Distribution Function) of 32 Hz data (grey-scale - legend in lower panel) and percentiles (yellow). (a) Mean value of $w'$ (green) and standard deviation of $w'$ (magenta) and Skewness of $w'$ (cyan). (b) Mean value of TKE from three component winds(solid green) and with only the vertical component (see text for details) dashed green).**

*Around Line 280, discussing estimate of INC based on aerosol larger than 0.5 um.*

*The authors provide an estimate based on DeMott et al (2015) and then appear to adjust (increase) that based on a correction factor of 3. But DeMott et als equation (2) includes CF (calibration factor as they call it). Thus to be consistent, authors should only report the larger value; the lower value of 0.3 L-1 doesnt make any sense, or at least is NOT CONSISTENT with DeMott et al (2015).*

We agree with the reviewer. Reference to uncorrected INP concentration is removed from the figure and the text. The formatting for the DeMott 2015 reference is also corrected.

**Resulting INP concentrations are $N_{INP}$ = 1.0 L$^{-1}$ (DeMott et al., 2015) and 0.60 L$^{-1}$ (Tobo et al., 2013).**

**2   Technical Comments**

*Line 4: change Turbulence data to Turbulence measurements*

Changed Turbulence data to Turbulence measurements

**Turbulence measurements are presented from both the liquid cloud layer and ice virga below**

*Line 16: . . .mixed-phase conditions began within a few metres. . .began is probably not the right work, it implies time. I sugges something like . . .mixed phase conditions were observed within a few metres of cloud top extending downward through the cloud*

We agree, began was not the correct word. Text has been altered.

**Carey et al. (2008) observed that mid-latitude altocumulus layer clouds were of mixed-phase composition on more than two-thirds of occasions and that mixed-phase conditions were observed within a few tens of metres of observable cloud top and extended down through the cloud**

*Line 89: abbreviation of knots should it be kts ??*

Abbreviation changed to kts

**During sampling the aircraft typically had a nominal Indicated Airspeed kts...**

*Line 275: Fig. 12@(b) ?*

typo corrected

**Fig. 12(b)**

A.A. Sinkevich and R.P. Lawson, A Survey of Temperature Measurements in Convective Clouds, J. Appl. Met. 44, 1133-1145, 2005
* * *
[Figure]

**Fig. 1.** Cloud particle number concentration histograms from CIP15 and CIP100

---

## Author Comment (AC4) · 7 Jan 2020

We thank the reviewer for the helpful comments and positive response to the paper.

The reviewer comments are in *italics*, author response in plain text, and modified manuscript text in **boldface**.

[Figure]

**1   Minor comments:**

*Abstract: The turbulence spectrum is observed to have an increasingly negative skewness with distance below cloud top, confirming that longwave radiative cooling from the liquid layer cloud is the source of turbulence kinetic energy. Personally, I would choose the more cautious term suggesting in lieu of the bolder term confirming. I have no doubt that cloud-top radiative cooling is relevant, but without a complete budget of TKE, it is hard to know the relative magnitude of each source.*

We agree that the data do not confirm turbulence generated through longwave radiative cooling and agree that "suggests" is more appropriate word for the abstract.

**The turbulence spectrum is observed to have an increasingly negative skewness with distance below cloud top, suggesting that longwave radiative cooling from the liquid layer cloud is an important source of turbulence kinetic energy.**

*Lines 40-41: The GCM simulations are found to have too little cloud in the mid-levels, resulting in a warm bias in sea surface temperatures, The authors might also want to mention Hartmann et al. (1992, J. Climate, see Fig. 22 and Table 1), which suggests that thin, mid-level clouds have little net radiative impact at the top of the atmosphere.*

The clouds in the study would indeed be of Type 3 (Hartmann 1992) as they are indeed optically thin, under this definition. We add a caveat and reference to the paper to state that while regional the clouds may have some radiative impact, the global significance of the cloud type is low.

**However, optically thin mid-level clouds were shown to be of low global significance by Hartmann et al. (1992).**

*Lines 74,75: which was predominantly from the south and ranged in strength from 6 m s-1 at the southern end of the flight track to 8 m s-1 in the north. It might also be worth recording the mean vertical wind shear within and near the cloud, because wind shear*

*is associated with generation of turbulence.*

We add comment that mentions the potential for turbulence generated through shear in these clouds.

**There was some degree of wind shear above the cloud containing layer with mean wind direction being close to northwesterly. This shear may have resulted in production of turbulence in the layer below.**

*Figure 8: On this figure or in its caption, please clarify what the diamond symbols mean, and how the reader is to infer whether the profiles are adiabatic by comparing to the many lines drawn on the figure. The description in Lines 218-224 is also a little unclear to me. E.g., a theoretical adiabatic ascent line is drawn starting at 136 m below cloud top. Was there clear, ascending air observed at this altitude? If not, what does the adiabatic line represent?*

Caption refers to Figure 4 for details of symbols. There are observations of updraughts in unsaturated air through the depth of the cloud system to as low as 500 m below the liquid cloud top. We now limit the plot to have 3 adiabatic ascents from three potential cloud bases. Cloud base was visually observed to be non-uniform, in the turbulence mixing- (but therefore not well-mixed) layer. This is supported by the Schmidt observations. The adiabatic ascents are shown to demonstrate that the in-cloud LWC distribution could arise as a result of this variable cloud base - which is in turn driven by the cloud top cooling generated turbulence, rather than by dilution through entrainment either at cloud top or laterally. The figure has been replotted to only show 3 ascents for clarity.

Caption 8 now reads: **LWC profile percentiles relative to CTH for the mixed-phase cloud layer with data from Nevzorov LWC sensor. Theoretical adiabatic ascents are shown, starting from three altitude (see legend for details) which correspond to potential cloud bases (see text for details). Statistics as Fig. 4**

The text has been ammended thus:

**Theoretical, undilute adiabatic, LWC profiles were calculated by assuming an ascent of a saturated air parcel from three initial altitudes corresponding to potential cloud bases. The first, from the minimum liquid cloud base at -223 m show that peak observed cloud top LWC values compare well with this theoretical estimate. An ascent from -167 m peaks close to the $75^{th}$ percentile of cloiud top LWC. A third ascent from -97 m has a peak LWC close to the $50^{th}$ pecenentile at cloud top. Whilst entrainment of dry air from aloft at the inversion may be non-aero, these calculations demonstrate that he non-uniform cloud base may have contributed to the observed in-cloud variability in LWC at a given level. This suggests a range of turbulent eddies and updraught depths contributed to the overall spectrum of in-cloud liquid water contents.**

*Fig. 10b: Are the habits of the ice particles photographed by CPI so complex because of aggregation or instead polycrystal formation because of defects (grain boundaries) in the crystals? The text in Lines 256,ÂŰ259 seems to suggest both. Is there a way to distinguish the two?*

We have not been able to disentangle the effects of polycrystaline growth from the impact of ice particle aggregation in the CPI imagery. Shadow imaging probe data are also not able to distinguish between the two. It is likely that both processes are active in the cloud. Attempting to quantify the aggregation rate in the clouds based on particle size number concertation etc, was deemed to be too complex and uncertain and so beyond the scope of the work. It may impact the ice production rate estimate to some degree.

*Fig. 13: cloud formation through wind shear or gravity wave activity acting at a stable interface in potential temperature Is it possible that the cloud forms from mesoscale or synoptic-scale lifting rather than gravity waves, and that the cloud-top inversion forms later through cloud-top radiative cooling? Is any estimate of large-scale vertical velocity*

[Figure]

*available?*

Following on from the discussion regarding turbulence generation from cloud top cool-
ing in the abstract, we agree, that without a full turbulence budget it is not clear the
source of the turbulence, but we also suspect that the wind shear has produced some
component of the TKE. There is no estimate of large scale ascent (unless we resort
to a reanalysis product), but it is certain that the clouds were in the warm sector of
a mid-latitude cyclone, and were experience slow large scale ascent along isentropic
surfaces in that region. Mention of large scale ascent has been added to the Figure
13 caption, in line with elsewhere in the text. We suspect the whilst this would pro-
duce supersaturation at some stage, the likelihood is that the first liquid clouds were
produced when additional uplift from either shear induced wave activity at the top of
the warm sector, or gravity wave activity at the same surface resulting from convection
in the frontal zones, was present. This cloud would then cool to space and thus begin
to produce the turbulence that could maintain the liquid cloud layer, and lead on to ice
production.

We agree that the cooling may produce the inversion later on, and have amended the
text to say so.

**Supersaturation and hence liquid cloud formation in the mid-levels of the tro-
posphere may be achieved through large scale ascent with upwards air motion
acentuated through wind shear or gravity wave activity, or convective detrain-
ment at a stable interface (Rauber and Tokay, 1991)**